# On-surface synthesis of phthalocyanines with extended π-electron systems
Lukas J. Heuplick[1], Qitang Fan[1,2,3], Dmitriy A. Astvatsaturov[4], Tatiana V. Dubinina [4] &
J. Michael Gottfried [1] ✉

Expanded phthalocyanines are a promising class of materials for optoelectronic applications, owing to their unique properties and versatile metal coordination reactivity. The expansion of their π-electron systems and resulting red-shifted absorption are of particular interest for achieving broader applications. Here, we report the on-surface synthesis of metallo-phthalocyanines with extended electron systems and an open-chain polycyanine from *ortho*-dicarbonitrile precursors on Ag(111) and Au(111), studied by scanning tunneling microscopy (STM) and X-ray photoelectron spectroscopy (XPS). The larger 6,7-di(2-naphthyl)-2,3-naphthalenedicarbonitrile (NND) undergoes spontaneous cyclotetramerization on the Ag(111) surface forming the corresponding silver naphthalocyanines (Ag-NPc), contrasting previous reports where a partially aliphatic *ortho*-dicarbonitrile precursor formed polycyanine chains. In contrast, monolayers of the smaller 6,7-diphenyl-2,3-naphthalenedicarbonitrile (PND) form the corresponding naphthalocyanine only in the presence of co-adsorbed iron atoms (Fe-NPc). In the absence of iron, PND multilayers form polycyanine chains and Ag-NPc. NND and PND further differ in their reactivity due to the supramolecular behavior of their products. While the larger Ag-NPc aggregates to non-covalent one-dimensional ribbons, the smaller Fe-NPc forms an extended non-covalent two-dimensional network. Our study demonstrates the versatility of on-surface dinitrile tetramerization for the synthesis of π-extended cyclic phthalocyanines and their open-chain polycyanine counterparts.

Porphyrins and other tetrapyrroles are essential in living organisms, serving vital functions in biomolecules such as hemoglobin and chlorophyll[1]. In technical applications, they are widely used due to their unique (opto-)electronic and (electro-)catalytic properties[2,3]. Phthalocyanines (Pcs) are closely related to porphyrins, but do not occur in nature. They show an enlarged π-electron system due to benzo-annulation and have the bridging methine groups replaced with nitrogen. These changes also alter their (opto-)electronic properties. Various free-base and metallated phthalocyanines were investigated on surfaces[1,4,5] and protocols for the on-surface synthesis[6,7] of metallophthalocyanines (e.g. CuPc[8], FePc[9], AlPc[10]) by metalation of the free-base phthalocyanine (2HPc) were developed. An alternative method for the on-surface synthesis of phthalocyanines involves the tetramerization of phthalonitrile derivatives using metal-atom templates, resulting in phthalocyanine complexes of Fe[11,12], Cu[13], Mn[14–16] and Au[17]. With larger templating metal atoms such as Gd, the related cyclic pentamers

(superphthalocyanines) have been obtained[18]. The on-surface reaction of a non-planar *ortho*-dicarbonitrile compound (ADN, Fig. 1a) in the absence of templating metal atoms (other than the Ag atoms of the substrate) resulted in the corresponding open-chain polymers (polycyanine chains). The formation of the chains was associated with the non-planar structure of the precursor[18]. In this work, we expand on the naphthalocyanine (NPc) synthesis reported previously[18], using two different non-planar *ortho*-dicarbonitrile precursors. 6,7-Di(2-naphthyl)-2,3-naphthalenedicarbonitrile (NND) extends a naphthalene *ortho*-dinitrile by two *ortho*-naphthyl groups and is known to undergo tetracyclization in solution in the presence of zinc[19]. As illustrated in Fig. 1b, NND in the adsorbed state has four possible non-planar conformers with one tilted naphthyl unit, due to intramolecular steric hindrance. Upon annealing NND on Ag(111) the formation of regular phthalocyanines, presumably Ag-NPc, is seen. This demonstrates the possibility of cyclotetramerization in the absence of

[1]Department of Chemistry, University of Marburg, Hans-Meerwein-Str. 4, 35032 Marburg, Germany. [2]Hefei National Research Center for Physical Sciences at the Microscale and Synergetic Innovation Center of Quantum Information & Quantum Physics, New Cornerstone Science Laboratory, University of Science and Technology of China, Hefei, Anhui, 230026, China. [3]Hefei National Laboratory, University of Science and Technology of China, Hefei, Anhui, 230088, China. [4]Department of Chemistry, Lomonosov Moscow State University, 119991 Moscow, Russian Federation. ✉e-mail: gottfried@uni-marburg.de

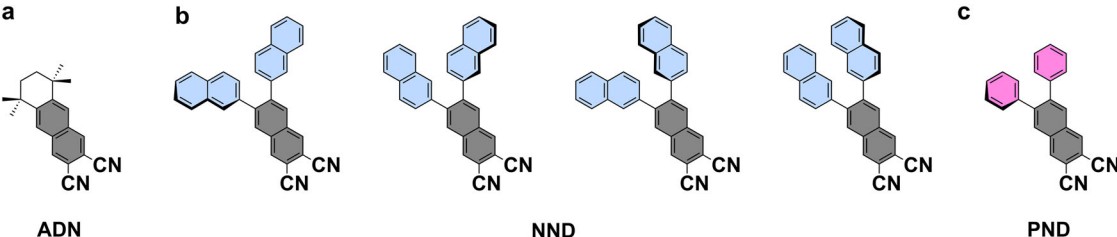

**Fig. 1 | Chemical structures of *ortho*-dinitrile precursors for naphthalocyanine synthesis. a** The 5,5,8,8-tetramethyl-5,6,7,8-tetrahydroanthracene-2,3-carbodinitile (ADN) precursor used previously[18]. **b** Four on-surface conformers of the 6,7-di(2-naphthyl)-2,3-naphthalenedicarbonitrile (NND) precursor. **c** 6,7-Diphenyl-2,3-naphthalenedicarbonitrile (PND) precursor utilized for the synthesis of Fe-NPcs, the polycyanine chain, and Ag-NPcs.

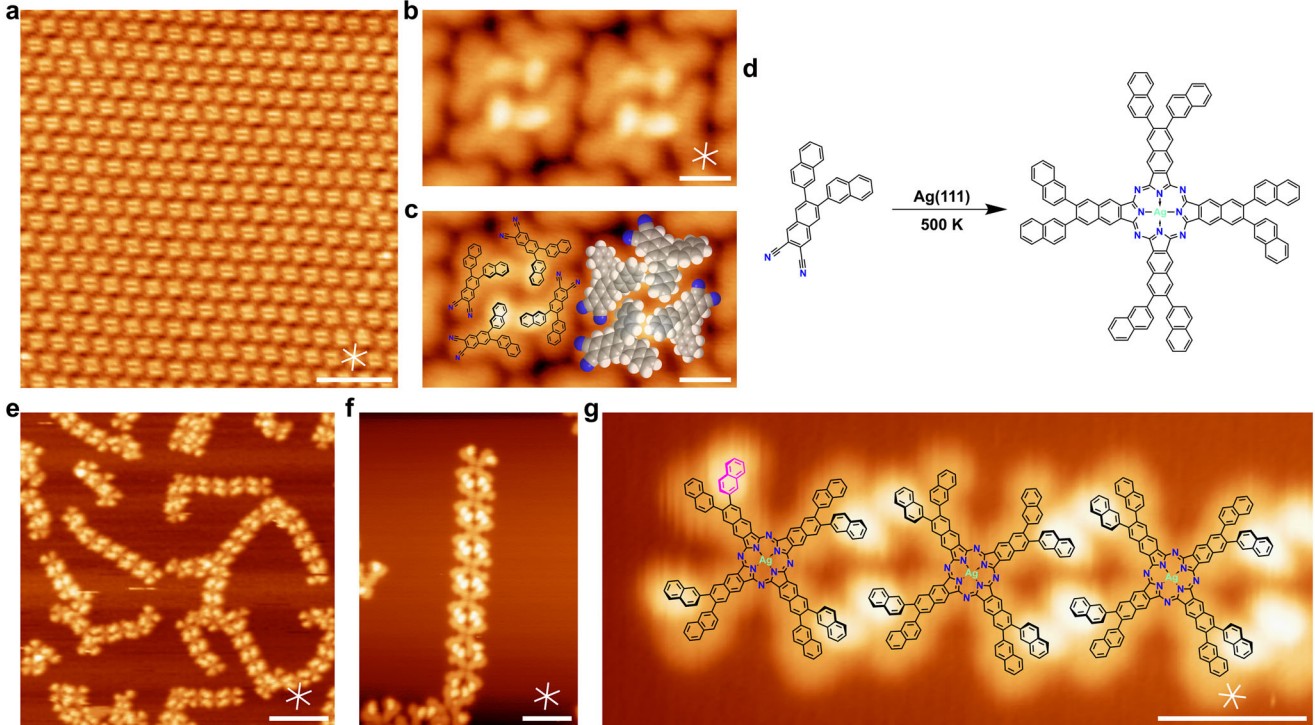

**Fig. 2 | Formation of Ag naphthalocyanine from the 6,7-di(2-naphthyl)-2,3-naphthalenedicarbonnitrile (NND) precursor on Ag(111). a** Large-scale image of a NND sub-monolayer (0.40 ML) on Ag(111) at 300 K. **b, c** Close-up image of the coordination of four precursor molecules to tetramer structures without (**b**) and with (**c**) overlaid molecular structures. **d** Reaction scheme of the formation of Ag-NPc upon annealing to 500 K. **e, f** Images of the ribbon-like arrangement of the Ag-NPc product after annealing to 500 K (300 K, 0.76 ML; 500 K, 0.40 ML). **g** Enlarged and rotated cutout of (**f**) with overlaid models, illustrating the connection of single Ag-NPc units to a ribbon via attractive intermolecular π-stacking. Highlighted in magenta is a single miss-oriented naphthyl-group, which prevents the continuation of the ribbon. Scale bars: (**a, e**) 10 nm; (**b, c**) 1 nm; (**f**) 4 nm; (**g**) 2 nm. Tunneling parameters: (**a**) $U = -2.67$ V, $I = -0.10$ nA; (**b, c**) $U = -2.93$ V, $I = -0.14$ nA; (**e**) $U = -2.75$ V, $I = -0.10$ nA; (**f, g**) $U = -2.75$ V, $I = -0.09$ nA.

co-adsorbed metal atoms and formation of naphthalocyanines with a further extended π-electron system. Due to the geometry of the additional naphthalene units, these regular M-NPcs form chain-like non-covalent supramolecular bands. The smaller 6,7-diphenyl-2,3-naphthalenedicarbonitrile (PND) closely resembles NND, but the naphthyl groups are replaced by phenyl substituents (cf. Fig. 1c). To achieve cyclotetramerization of the PND precursor at (sub-)monolayer coverages, co-adsorbed iron atoms are necessary, otherwise the precursor desorbs unreacted. However, using a sufficiently high coverage in the multilayer regime also induces reaction, notably resulting in the formation of polycyanine chains. The resulting M-NPc complexes arrange in a long-range ordered two-dimensional structure, while the polycyanine chains typically at least in pairs. For both precursors, cyclodehydrogenation and thus flattening of the substituents seems to be possible at higher annealing temperatures, resulting in a further extension of the π-electron system (see below for PND and Fig. S1a, b for NND).

## Results and Discussion

Deposition of 0.40 monolayers (ML) of NND onto Ag(111) at 300 K yields long-range ordered areas of molecules arranged in non-covalent tetramers, as shown in Fig. 2a (unit cell: 2.93 nm × 2.80 nm, 119°). A zoom-in image (Fig. 2b) shows that the tetramers are composed of four individual y-shaped features, in which one lobe appears brighter. This agrees well with the expected geometry of a single adsorbed NND molecule. As indicated in a molecular overlay (Fig. 2c), the bright protrusions correspond to tilted naphthyl substituents. This tilting is typically found for compounds with similarly rotationally flexible carbon-carbon single bonds in the literature and leads to an increased apparent height; examples include 5,10,15,20-tetraphenylporphyrin[20] or poly(para-phenylene)[21]. This overlay further helps identifying the driving force of the ordering. The tetramers are likely stabilized by attractive interactions between the π-electron systems of tilted naphthyl groups of neighboring molecules, whereas neighboring tetramers appear to be connected by C-H⋯N hydrogen bonds.

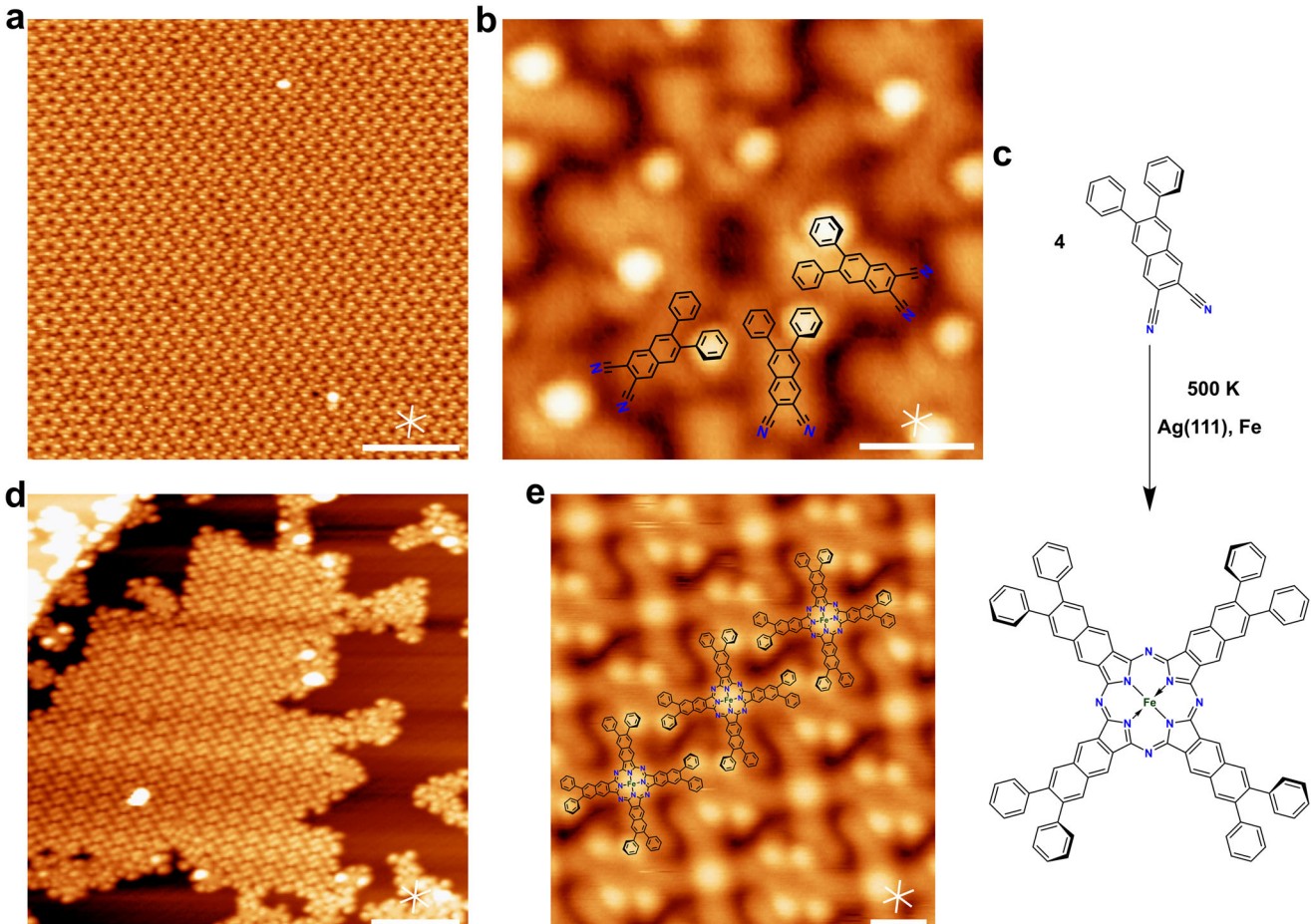

**Fig. 3 | Formation of Fe naphthalocyanine from the 6,7-diphenyl-2,3-naphtha-lenedicarbonitrile (PND) precursor and co-adsorbed Fe on Ag(111). a** Large-scale image of a PND sub-monolayer (0.35 ML) on Ag(111) at 300 K. **b** Close-up image with overlaid molecular model, showing the formation of supramolecular hexamers. **c** Reaction scheme for the formation of Fe-NPc on Ag(111) upon annealing to 500 K (300 K, 0.52 ML; 500 K, 0.33 ML) in presence of co-adsorbed Fe. **d** Island formation of regular Fe-NPc molecules after annealing to 500 K. Irregular shaped structures lead to unordered fractions around the islands. **e** Close-up image of an island as in (**d**) with an overlaid model, which shows the staggered structure of the Fe-NPcs, stabilized by attractive π-π-interactions. Scale bars: (**a**, **d**) 10 nm; (**b**, **e**) 1 nm. Tunneling parameters: (**a**) $U = -1.68$ V, $I = -0.11$ nA; (**b**) $U = -2.22$ V, $I = -0.36$ nA; (**d**) $U = -2.93$ V, $I = -0.08$ nA; (**e**) $U = -2.59$ V, $I = -0.14$ nA.

After annealing to 500 K, the ordered islands have disappeared and ribbons of varying length and orientation have formed, as shown in Fig. 2e. The close-up image in Fig. 2f reveals that the ribbons consist of cross-shaped features, which have dark centers. In addition, each of the four lobes of a cross have a brighter and a darker part. In line with previous work[11–18], we attribute the cross-shaped features to regular naphthalocyanines formed by the cyclotetramerization of NND with Ag-adatoms to Ag-NPc (Fig. 2d). Spectroscopic evidence for the formation of Ag-NPc will be discussed below. A molecular overlay of the enlarged ribbon in Fig. 2g illustrates the non-covalent intermolecular connection. It is proposed that π-π-interactions due to the tilting of the naphthyl substituents lead to the formation of these non-covalent ribbons.

The ribbons are relatively short and their straight sections rarely exceed 10 repeat units. A possible explanation for the termination of a ribbon can be deduced from a closer inspection of the intramolecular structure of the terminal Ag-NPc molecules. While all possible conformations of the two naphthyl groups (cf. Fig. 1b) appear in the final products and contribute to the propagation of the ribbons, only those molecules whose "inner" naphthyl groups (with respect to the ribbon orientation) are tilted can continue the ribbon. Therefore, if an "outside" group is upright (magenta colored group in Fig. 2g), the ribbon terminates. For further explanations regarding the contributions of each conformer, see Fig. S2.

The STM images alone do not provide sufficient evidence for the presence of a central metal atom in the Ag-NPc, because the center of the molecule appears dark. This is in line with previous work for Ag phthalocyanine, which usually adopts a "Ag down" conformation in the adsorbed state, where the silver atom is below the molecular plane[22]. However, the N 1 s XPS spectra discussed below clearly prove the absence of aminic N-H, which is typical for free-base phthalocyanine. This strongly suggests formation of a metallophthalocyanine complex, which, in the absence of other metals, can only be a Ag complex. Furthermore, mechanistic studies of the self-metalation of phthalocyanines on Ag surfaces show that the formation of AgPc from the N-dehydrogenated Pc with adatoms is energetically favored[23]. Therefore, we can also exclude that the center of the formed NPc is only N-dehydrogenated, but not metallated.

After deposition of the smaller PND precursor on Ag(111), extended islands of distinguishable features ordered in a nearly hexagonal lattice (unit cell: 2.57 nm × 3.02 nm, 116°) are observed (Fig. 3a). A close up (Fig. 3b) reveals that each feature is composed of 6 individual molecules. Similar to those in Fig. 2b, they appear in an asymmetric y-shape, containing one brighter lobe. This agrees well with the molecular structure of PND in the adsorbed state, as indicated by the structural overlay. As for NND, the high degree of order of the non-covalent hexamers is attributed to the π-π-interactions of the phenyl groups, whereas the quasi-hexagonal super-structure appears to be stabilized by hydrogen-bond interactions between the cyano-groups and the neighboring hydrocarbon backbone. Annealing a sub-monolayer coverage (0.7 ML) of PND to 500 K results in desorption without formation of phthalocyanines or other oligomers. In contrast,

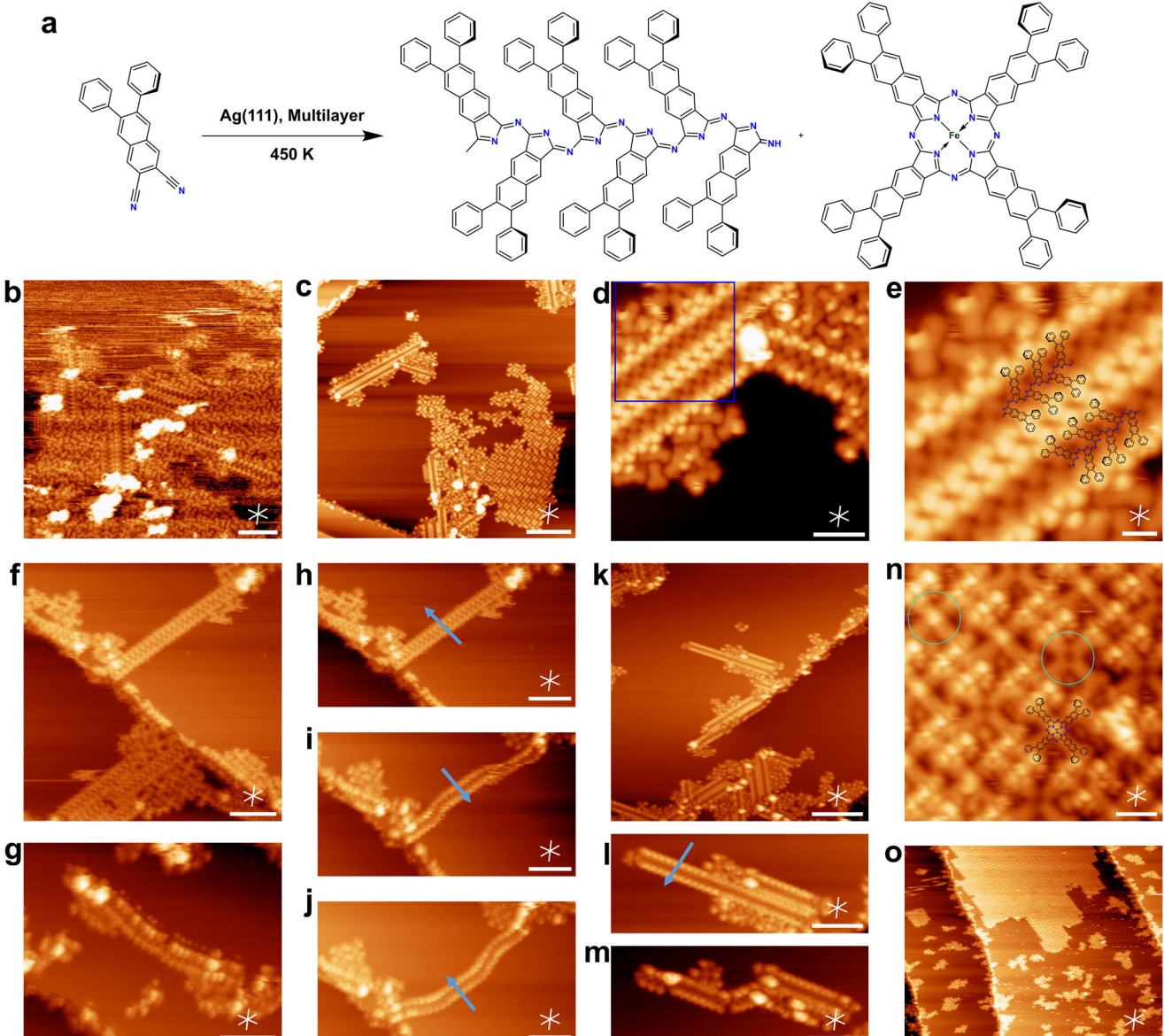

**Fig. 4 | Formation of polycyanine chains in multilayers of the PND precursor on Ag(111). a** Reaction scheme for the formation of polycyanine chains and Ag-NPc cyclic covalent tetramers in PND multilayers at 450 K. **b** Large-scale STM image taken after annealing of a PND multilayer (~8 × 10^14 molecules/cm²) to 450 K on Ag(111). **c** STM image showing that further annealing of the sample in (**b**) to 470 K leads to desorption of residual monomers and the coexistence of chains and cyclic tetramers. **d** Close-up STM image of the polycyanine chains in (**c**). **e** Zoom-in STM image as indicated in (**d**) with overlaid chemical model. **f** Single chain perpendicular to a step edge before and parallel to the edge after (**g**) repeated lateral tip manipulations along the directions of the arrows in (**h–j**). **k** Different ensemble of chains. **l** Close-up STM image of the chain in (**k**) before manipulation. **m** Chain after tip

manipulation showing a dent and additional bright protrusions attributed to upstanding chain segment. **n** Close-up STM image of the tetramer island in (**c**), showing cross-shaped cyclic covalent tetramers (NPcs) with and without bright central protrusions. **o** Large-scale STM image of the sample upon annealing to 550 K. Scale bars: (**b, f–j, m**) 5 nm; (**c, k**) 10 nm; (**d**) 3 nm; (**e, n**) 1 nm (**o**) 20 nm. Tunneling parameters: (**b**) $U = 1.20$ V, $I = 0.11$ nA; (**c**) $U = -1.28$ V, $I = -0.26$ nA; (**d, e**) $U = -1.40$ V, $I = -0.24$ nA; (**f–i**) $U = -2.59$ V, $I = -0.19$ nA; (**j**) $U = -3.02$ V, $I = -0.21$ nA; (**k**) $U = -2.15$ V, $I = -0.20$ nA; (**l**) $U = -2.15$ V, $I = -0.18$ nA; (**m**) $U = -2.15$ V, $I = -0.15$ nA; (**n**) $U = -0.20$ V, $I = -0.28$ nA; (**o**) $U = -3.31$ V, $I = -0.13$ nA.

annealing of PND multilayers results in the formation of covalent polycyanine chains (cf. Fig. 4a and text below).

In the presence of co-adsorbed iron (here sub-stoichiometric amount as determined by XPS; higher ratios on Ag(111) as well as on Au(111) are discussed in Fig. S3), however, annealing of a PND sub-monolayer (0.52 ML) leads to the formation of the corresponding iron naphthalocyanine (Fe-NPc) complexes, according to the reaction scheme shown in Fig. 3c. As can be seen in Fig. 3d, the complexes form islands with a nearly quadratic unit cell (2.25 nm × 2.10 nm, 91.3°). A close-up of the ordered islands reveals cross-shaped molecules with a bright Fe center, in agreement

with previous work about adsorbed Fe phthalocyanines[9]. Each of the four lobes of the molecule shows one bright protrusion, which is attributed to one rotated phenyl ring. As illustrated by a scaled molecular overlay in Fig. 3e, this agrees well with the expected shape of Fe-NPc. For comparison, we also studied the reaction of NND on Ag(111) with co-adsorbed iron (see Fig. S1c, d). However, in contrast to the ordered structures obtained with PND, mainly irregular cyclic oligomers were formed upon annealing to 500 K (Fig. S1d).

Annealing of a PND multilayer (~8 × 10^14 molecules/cm²) on Ag(111) to 450 K without prior exposure to X-rays (cf. Supplementary Fig. S4)

**Fig. 5 | XPS spectra of the NND and PND precursors and the corresponding naphthalocyanine products on Ag(111).** N 1 s and C 1s XPS spectra of both precursors (**a, e, c, g**) and their corresponding M-NPcs (**b, f, d, h**) on Ag(111). All spectra were taken at 300 K. The N 1 s data show the conversion of the cyano groups to the phthalocyanines by a shift towards lower $E_B$ and peak broadening. The C 1 s data (**e, g**) were fitted by taking the stochiometric factors of the three different main species (nitrile, tilted and non-special carbon) in account. The products (**f, h**) show a decreased intensity of the shoulder arising from the contribution of the nitrile groups (blue).

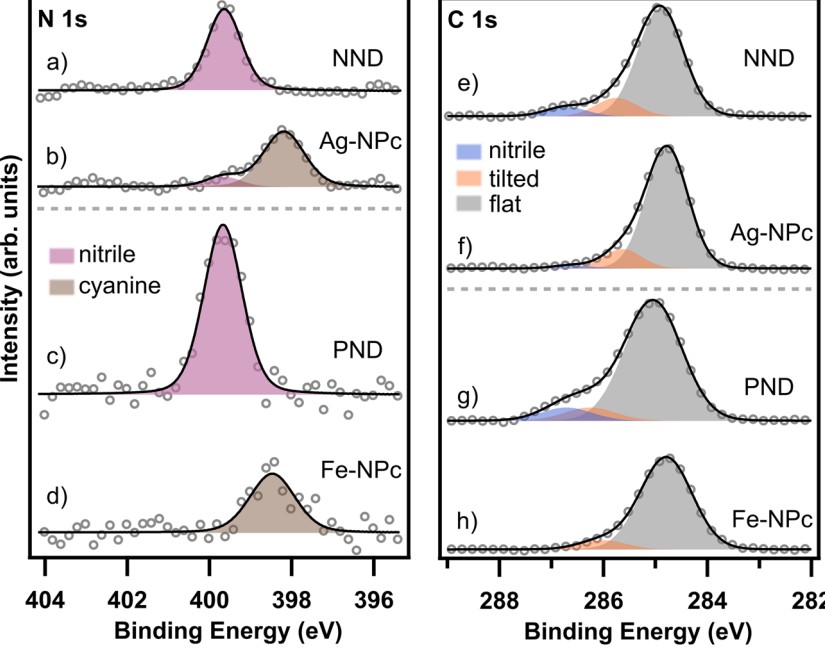

resulted in the formation of extended linear features, which are attributed to polycyanine chains (see the structure in Fig. 4a), in line with previous work[18]. These chains are surrounded by smaller moieties (Fig. 4b), which are most likely unreacted PND molecules. To desorb the residual unreacted PND, the sample was annealed to 470 K, resulting in a decrease of the total coverage from 0.52 ML to 0.39 ML, according to XPS. The STM image in Fig. 4c, taken after the annealing, shows linear chains coexisting with islands containing cross-shaped fourfold symmetric molecules, which will be discussed later. Closer inspection of the chains reveals that each chain consists of three rows, a dark row in the center and two bright rows along the edges (Fig. 4d, e). This matches well with the expected appearance of a polycyanine chain[18], where the inner N-containing part binds more strongly to the substrate, whereas the edges are bent upward due to the non-planarity of the aromatic rings system. The chains agglomerate, as can be seen in Fig. 4d, e, which is attributed to attractive interactions between the phenyl-substituents at the edges. To further confirm the formation of the covalent polycyanine chain (as opposed to a metal-organic or van-der Waals linked chains (cf. Fig. S4 for further details)), lateral tip manipulation of two different chains was performed (Fig. 4f and k). By moving the tip along the indicated arrows in Fig. 4h–j and l, we were able to bend an individual chain multiple times without breakage (Fig. 4h–j), before a 90° rotation of the complete chains and further deformation of the chain occurred. By applying similar lateral movements to another, partly embedded chain (Fig. 4l), bright protrusions and small dents appear at the bending positions. These protrusions were also seen previously for another polycyanine chain[18] and was attributed to upright-standing chains segments due to compression at one edge of the chain.

The zoom-in STM image (Fig. 4n) of the cross-shaped product reveals two distinct types of molecules, with and without a bright protrusion in the center. In line with the discussion for the NND cyclotetramerization above, we attribute these products to the regular M-NPcs, which, in the absence of other co-adsorbed metals, are assumed to incorporate Ag adatoms. The different contrasts at the centers are attributed to two different configurations, in which the Ag atom is either below the molecular plane (Ag-down, dark center, as in Fig. 2) or above the molecular plane (Ag-up, bright protrusion), in agreement with previous work on Ag phthalocyanine on Ag(111)[22]. We propose that the presence of the organic multilayer during the formation of the covalent cyclic tetramers facilitates flipping movements of the Ag-NPc complex, resulting in a mixed phase of Ag-up and Ag down products.

The mixed phase of PND-based polycyanine chains and Ag-NPc was annealed to 550 K for 15 min to induce possible cyclodehydrogenation reactions of the phenyl substituents. After annealing, however, no chain-like structures were found on the surface (Fig. 4o). This change is not due to desorption, as there was only a marginal reduction of the total coverage (from 0.39 ML to 0.36 ML, according to XPS). Instead, the coverage of Ag-NPc increased (cf. Fig. S4f and g for a close-up image), indicating a follow-up degradation reaction of the polycyanine chains, resulting in the formation of Ag-NPc. This observed transformation suggests that the Ag-NPc represents the thermodynamically more stable product, while the polycyanine chains are the kinetically preferred product, in agreement with previous work on the ring-chain competition in on-surface oligomerizations[24].

Further evidence for the formation of naphthalocyanines by cyclotetramerization is provided by the XPS spectra in Fig. 5. For NND and PND monolayers on Ag(111), the N 1 s XPS spectra show distinct peaks at binding energies ($E_B$) of 399.6 eV (Fig. 5a) and 399.7 eV (Fig. 5c), respectively. Both agree well with findings for cyano-groups in similar environments (e.g. 399.6 eV in previous work[18]). Upon annealing of the NND layer, a second peak appears at 398.2 eV, while the intensity of the original peak decreases. This agrees with findings for Ag-Pc in previous work (398.3 eV[23]). Note that the small binding energy difference between the inner nitrogen atoms and the meso-nitrogen atoms[25,26] is not resolved here, but leads to increased width (+ 20%) of the N 1 s peak. Similarly, annealing of PND to 500 K in presence of iron leads to a new N 1 s peak at 398.5 eV, while the peak at 399.7 eV shrinks (Fig. 5c, d). This confirms the proposed cyclotetramerization of PND to Fe-NPc. While the observed shift of −1.2 eV is less than that reported by Fan et al. (−1.7 eV[18]), the binding energy of regular iron-phthalocyanine (398.7 eV[9]) is only 0.2 eV higher than we found for Fe-NPc. Fan et al. report a binding energy of 397.9 eV[18] and attribute this difference to the stronger nitrogen-substrate interaction due to the non-planarity of their peripheral substituent. Reduced non-planarity of the tilted phenyl-group in PND, therefore, might explain the binding energy shift. The C 1 s data of the precursors show a distinct peak with a shoulder at higher binding energies and were fitted with three distinct contributions: the base structure (gray, NND: 284.9 eV; PND: 285.0 eV), the C atoms in the upstanding aryl substituents (orange, NND: +0.8 eV; PND: +1.2 eV), and the carbon atoms of the nitrile-groups (blue, NND: +1.8 eV, PND: +1.7 eV; cf. ADN: +1.6 eV[18]). The C 1 s fits in Fig. 5e and g were obtained by using the expected stoichiometric ratios (NND 2:4:26; PND 2:2:20). Upon reaction, the contribution of the cyano-groups decreases (Fig. 5f) or disappears

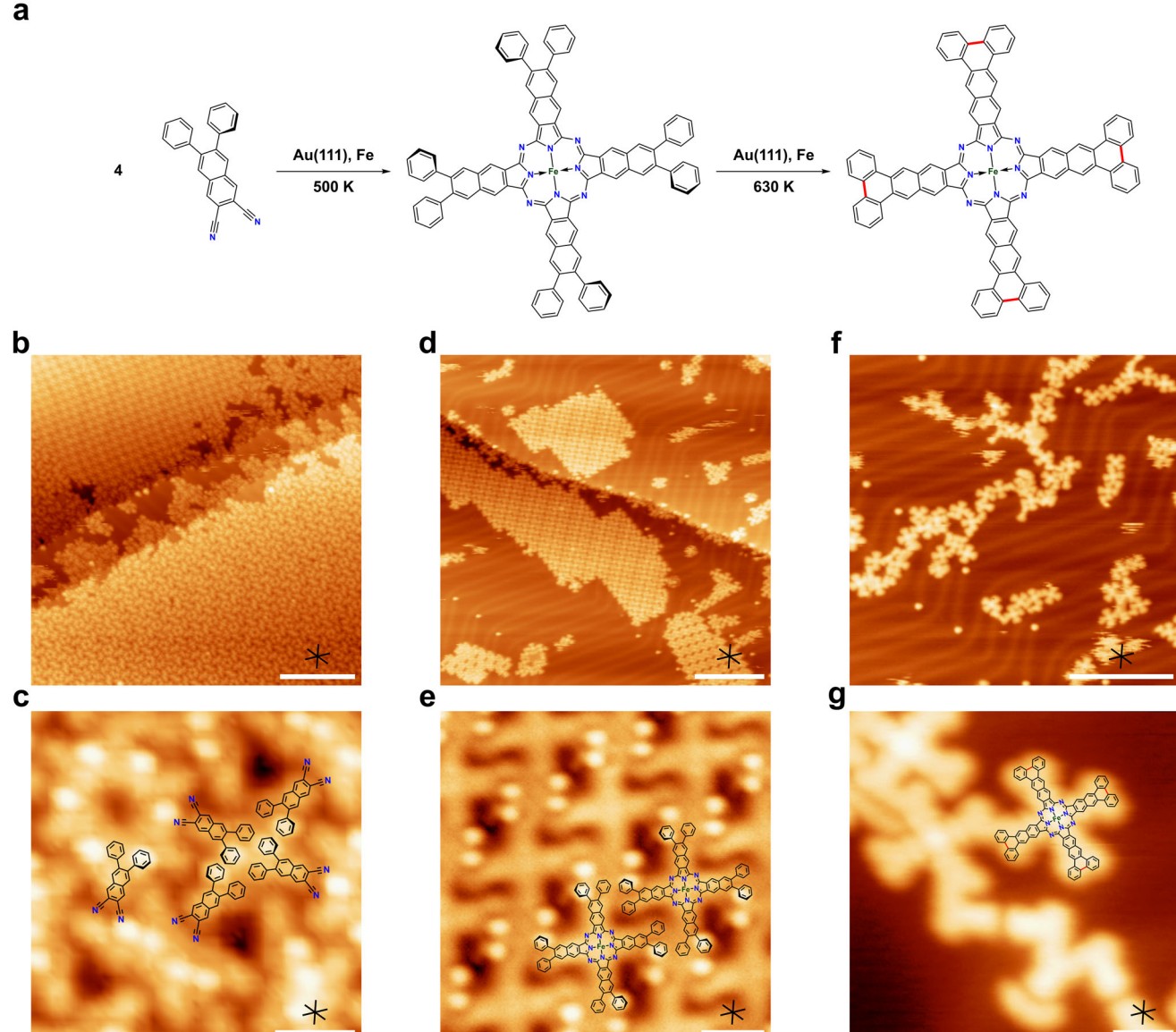

**Fig. 6 | Formation of Fe naphthalocyanine from the PND precursor and co-adsorbed Fe on Au(111). a** Proposed reaction scheme for PND with Fe on Au(111). **b** STM image of PND on Au(111) at 300 K (0.61 ML). Two different domains can be distinguished in the adsorbate structure. Upper left: Structure similar to NND on Ag(111) (cf. Fig. S8 for further details); lower right: mixed phase of tetramers and hexamers. **c** Close-up STM image of the mixed phase from (**b**) with overlaid model, which shows the coordination of four PND molecules to non-covalent tetramers. **d** STM image after annealing to 500 K showing the formation of islands with straight edges consisting of regular Fe-NPc molecules (coverages: 300 K, 0.67 ML; 500 K,

0.34 ML). **e** Close-up STM image of an island as in (**d**) with an overlaid model, showing the regular pattern of the Fe-NPcs, stabilized by attractive π-π-interactions. **f** Larger-scale STM image of the sample after annealing to 630 K (0.26 ML). **g** STM image of a fully planarized Fe-NPc molecule next to non-regular structures. Scale bars: (**b**, **d**, **f**) 20 nm; (**c**, **e**, **g**) 1.5 nm. Tunneling parameters: (**b**) $U = -1.00$ V, $I = -0.21$ nA; (**c**) $U = 2.02$ V, $I = 0.09$ nA; (**d**) $U = -1.85$ V, $I = -0.13$ nA; (**e**) $U = 1.44$ V, $I = 0.12$ nA; (**f**) $U = -2.67$ V, $I = -0.17$ nA; (**g**) $U = -2.84$ V, $I = -0.13$ nA.

(Fig. 5h), and the main peak increases relatively, which is in line with the incomplete reaction of NND to Ag-NPc already shown in the N 1 s region. Further C1s data of other temperatures are shown in Fig. S5.

To explore possible influences of the metal substrate on the reaction, the reactive cyclotetramerization of PND was also studied on the Au(111) surface. After deposition at 300 K (Fig. 6b), a different adsorption geometry compared to Ag(111) was observed. Instead of the extended superstructure of non-covalent hexamers, we find domains with at least two different structures in approximately similar abundance: a mixed phase of non-covalent hexamers and tetramers (Fig. 6b bottom right, Fig. 6c, unit cell 5.52 nm × 5.72 nm, 117.6°) and a phase with only non-covalent tetramers similar to NND (Fig. 6b top left, unit cell 2.34 nm × 2.36 nm, 92.1°). For further details about of the coordination network without and with co-

adsorbed Fe at 300 K, see Fig. S8. Upon annealing of PND on Au(111) to 500 K in the presence of Fe, ordered islands are observed (Fig. 6d, unit cell 2.43 nm × 2.29 nm, 101°), which reveal cross-shaped molecules in the zoom-in view (Fig. 6e). As before, we assign these product molecules to the corresponding Fe-NPc resulting from the cyclotetramerization reaction in Fig. 6a. To exclude the formation of Au-NPc, we also annealed a pristine sample of PND (0.61 ML) on Au(111) as well, resulting predominantly in desorption without cyclotetramerization (Fig. S6). Note that the edges of the islands are straighter than on the Ag(111) surface. This can be explained by differences in the adsorbate-substrate interaction. Due to the weaker interaction with the Au(111) surface, the intermolecular interactions are comparatively more significant. Deviations from the edge structure with the lowest energy (i.e., straight edges) can therefore be stabilized on Ag(111), but

less so on Au(111), resulting in more well-defined edges on Au(111). To investigate whether the larger NND precursor would show similar trends, we performed additional experiments with this precursor on Au(111). The data in Fig. S7 show that the reactive cyclotetramerization on Au(111) has a very low yield and only few cross-shaped M-NPc products with dark and bright centers were observed, which may be attributed to Au-NPc in the Au-down and Au-up state, respectively (see especially Fig. S7f and h).

Annealing of the PND/Fe co-adsorbate on Au(111) to even higher temperatures up to 630 K resulted in a loss of the long-range order (Fig. 6f). The close-up image in Fig. 6g reveals that some cross shaped molecules are still present. Their lobes show uniform contrast without the bright protrusions, indicating that intramolecular cyclodehydrogenation and C-C coupling have occurred, resulting in a planarization of the molecules. The planar molecules experience less lateral intermolecular stabilization in the islands due to π-π-interactions, which explains the observed lack of long-range order. The flattening is also expected to extend the π-conjugation in the whole macrocycle and thereby to influence its optoelectronic properties. Besides the intramolecular cyclodehydrogenation, the molecules also partly engage in intermolecular C-C coupling, which is evident from the representative image in Fig. 6g.

## Conclusions

We have shown the on-surface synthesis of extended naphthalocyanine derivatives by metal-template directed reactive cyclotetramerization of the *ortho*-naphthalodinitriles NND and PND, as well as the metal-template assisted formation of a polycyanine chain from PND. Annealing of NND on Ag(111) to 500 K resulted in regular naphthalocyanines with extended π-electron systems incorporating a silver atom from the substrate. These Ag-NPc molecules form one-dimensional supramolecular ribbons due to their intrinsic adsorption geometry, which leads to stabilizing π-π-interactions of their tilted naphthyl substituents. Multilayers of PND on Ag(111) at 450 K were found to form the open-chain covalent polycyanine, which are converted to regular naphthalocyanines at 550 K. No similar reactions were observed for sub-monolayer coverages of PND or for NND. While sub-monolayers of PND were found to desorb on both Au(111) and Ag(111) without engaging in reactions, the reactive cyclotetramerization of sub-monolayer PND was induced by co-adsorbed Fe atoms. The resulting Fe-NPcs form two-dimensional islands. Cyclodehydrogenation upon annealing of FeNPc on Au(111) to 630 K resulted in a fully conjugated flat macrocycle. Our results show that the on-surface cyclotetramerization reaction of aromatic *ortho*-dicarbonitriles (naphthalodinitriles) is a versatile approach for preparing layers of large benzo-annulated naphthalocyanines with extended π-electron systems. Further characterization of their optical properties is expected to reveal a red-shifted adsorption, contributing to a broader range of optically active surface structures.

## Methods

All experiments were performed under ultra-high vacuum (UHV) conditions with a base pressure below $5 \times 10^{-10}$ mbar in a combined STM-XPS apparatus from SPECS. All STM data were measured using a SPECS STM Aarhus 150 at temperatures below 120 K in constant current mode. The bias voltages are referenced to the sample. Moderate flattening in form of background subtraction was applied in WSxM[27], all close-up STM images were corrected to compensate thermal drift effects and all given crystal orientations are derived from substrate signals (cf. Fig. S9). Molecular overlays were scaled according to calculated atomic distances (cf. Supplementary Information, Fig. S10). XPS data were acquired using a SPECS PHOIBOS 150 electron analyzer equipped with an MCD 9 detector. The XPS spectra are referenced to the Ag 3d$_{5/2}$ peak at 368.3 eV[28] and are shown after background subtraction. All given coverages are – in absence of a measured saturation coverage - referenced to the C 1s signal of one saturated layer (1 monolayer; 1 ML) of 5,10,15,20-tetraphenylporphyrin on the respective substrate. The Ag(111) and Au(111) single crystals were obtained from MaTecK with a surface misalignment of no more than 0.1° and a surface roughness of <0.01 μm. Prior to sample preparation, the crystals were purified by cyclic Ar$^+$ ion bombardment (0.5 keV) for 15 min and flash annealing to 800 K. NND and PND were evaporated from homemade Knudsen cells at 518 K (NND), 455 K (PND), 432 K (PND multilayer) onto the crystal held at 300 K. Fe was deposited using an EFM4 electron beam evaporator at a flux of 1.3 nA for 45 s (Ag(111)) or 60 s (Au(111)). The indicated annealing temperatures were kept for 15 min or 3 min (PND multilayer experiments). The XPS data of NND were acquired from a comparable sample prepared with a similar coverage, but annealed to 500 K in two steps for 15 min and 30 min, respectively.

## Supporting Information

Synthesis and characterization of 6,7-di(2-naphthyl)-2,3-naphthalenedicarbonitrile and 6,7-Diphenyl-2,3-naphthalenedicarbonitrile. Cyclodehydrogenation and experiments with co-adsorbed iron of NND. Analysis of the different conformers. Changed reactivity upon increased iron coverage. Further C 1s XPS data. Desorption of PND on Au(111). Reactivity of NND on Au(111). Additional STM data of PND on Au(111). Determinations of the crystal orientations. Size adjustments of molecular overlays.

## Data availability

The source data that support the findings of this study are available from the corresponding author upon reasonable request.

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

## Acknowledgements
Financial support by the Deutsche Forschungsgemeinschaft (DFG, German Research Foundation) through grants 223848855-SFB 1083 and GO1812/4-1 and by the State of Hessen through the LOEWE Focus Group PriOSS is gratefully acknowledged. This project was supported by the European Regional Development Fund (ERDF) and the Recovery Assistance for Cohesion and the Territories of Europe (REACT-EU).

## Author contributions
J.M.G., Q.F. and T.V.D. planned this project. D.A.A. and T.V.D. synthesized and characterized the precursor molecules. L.J.H. and Q.F. performed the STM and XPS measurements of PND. L.J.H. and Q.F. performed the STM measurements of NND. L.J.H. conducted the XPS experiments of NND. L.J.H. analyzed all data. J.M.G. and L.J.H. wrote the draft of the manuscript. All authors discussed the results and commented on the manuscript.

## Funding

## Competing interests
The authors declare no competing interests.
