## [Peer Review File · Communications Chemistry]

On-Surface Synthesis of Phthalocyanines with Extended π -Electron Systems

Corresponding Author: Professor J. Michael Gottfried

Version 0:

Reviewer comments:

Reviewer #1

(Remarks to the Author)

The paper is recommended for publication, but minor revision is advised.

The authors describe temperature-induced on-surface synthetic cyclotetramerization and metalation reactions of NND and PND towards metal-phthalocyanines, on noble metal (111) surfaces (Ag and Au, sometimes including coadsorbed Fe atoms) studied by STM and XPS. In both cases, thermal annealing to 500 K has been applied to induce the process. After these annealing steps, seemingly only the yield of finalized molecules can be observed on the surface. Intermediate products as well as non-reacted precursors have been desorbed during annealing. Therefor arises the question whether shorter annealing times or somewhat lower annealing temperatures could possibly allow some further insight into the details of the ongoing reactions.

Further, does the amount of coadsorbed Fe atoms determine the yield of the metal-naphthalocyanines? Otherwise, one could expect to find single excess Fe atoms.

NND is expected to exhibit two isomers for each of the three conformers (Fig. 1) on the surface plane (having the tilted unit either on the left or the right side, respectively). For PND similarly two surface isomers are expected. Nevertheless, only one kind of them contributes to the final molecule (except for rarely encountered defects as shown in Fig. 2g). Are there observed complete mirror-counterparts in other regions of the substrate, or might the precursors molecules change mutually into the other isomer during annealing (e.g. by simultaneous hopping and twisting)?

In Figures 2, 3, and 5, the spatial orientation of the substrate surface should be indicated.

There is shortly mentioned an additional phase visible in Figure 5b, top left. A close-up STM image of this phase should be added if possible (at least to the supplementary information).

In the references, there is sometimes a complete listing of all the authors otherwise an abbreviated list using "et al". I would prefer to have always the complete list for "et al." discriminates some of the coauthors, in my opinion.

Reviewer #2

(Remarks to the Author)

Expanded phthalocyanines are a promising class of materials for optoelectronic applications. In this manuscript, the authors report the on-surface synthesis of two metallo-phthalocyanines Ag-NPc and Fe-NPc with extended electron systems from two different ortho-dicarbonitrile precursors studied by STM and XPS. The experimental results are interesting and important. The interpretation seem reasonable. The manuscript is well organized and written. However, there still are a few places that are not clear, which are listed below. If the authors could reasonably address these questions and comments, I would support the publication of this manuscript in the Communication chemistry.

1. Both the tetrameric structure of NND molecules on the Ag(111) surface and the formation of Ag-NPC reveal two benzene molecules that appear brighter. The author attributes this to the tilt of the groups. Why do they appear brighter after tilting? Is it due to higher spatial or charge effects?

2. The author did not provide a coordination assembly structure induced by silver atoms from the assembly structure of NND molecules on the Ag(111) surface to the formation of the final polymer. Has the author not observed this coordination structure, or is it indeed not formed?

3. Reference 17 documented the creation of AuPC on the Au(111) surface using cyanide groups and surface Au atoms, tracing the detailed evolution from assembly structure to coordination structure, culminating in the final AuPc product. How did the author ascertain that the entity depicted in Figure 5 is Fe-NPC, considering it could potentially be Au-NPC? Has the author experimented with directly heating PND molecules on the Au(111) surface to explore the synthesis of AuNPC? Given that this paper primarily delves into investigating the impact of molecular configuration, substrate, and metal reactivity on phthalocyanine polymer synthesis, it should incorporate experimental data regarding NND on Au(111).

4. The "NND" on line 130 of page four should be "PND".

Reviewer #3

(Remarks to the Author)

The manuscript of Lukas J. Heuplick and co-workers reports the on surface synthesis and characterization of two metallo-phthalocyanines (M-Pcs) on Ag(111) and on Au(111), a topic where the authors have expertise and relevant previous publications. In this work they report the cyclotetramerization of 6,7-25 di(2-naphthyl)-2,3-naphthalenedicarbonitrile (NND) on Ag(111) leading to the silver naphthalocyanine, and the synthesis of a smaller iron naphthalocyanine on Au(111) and Ag(111) by cyclotetramerization of 6,7-diphenyl-2,3-naphthalenedicarbonitrile (PND) in the presence of co-adsorbed iron atoms. Throughout the manuscript, the authors discuss the formation of these metallo-phthalocyanines and their supramolecular arrangements supporting their claims with scanning tunnelling microscopy (STM) and X-Ray photoelectron spectroscopy (XPS) data. I consider the manuscript of interest to the field and I would recommend the work of Lukas J. Heuplick et al. to be published in Communications Chemistry after addressing the following points.

- 1) In the line 76, it could be good to include the estimated number of ML deposited on Ag(111). Figure 2a suggests a complete ML, is that the case?
- 2) In Figure 1 and throughout the manuscript, it is discussed that different conformers of the NND precursor coexist when adsorbed on a surface, but there is no mention of the stability of these three conformers on Ag(111). As it is the main reason for the termination of the non-covalent ribbons of Ag-Pcs, it could be interesting to include how many different conformers appear on the surface and an estimation of the % of each one after depositing NND precursor on Ag(111) and after the annealing up to 500K.
- 3) About Ag-Pcs synthesised from NND precursor: Have the authors tried to anneal above 500K? Does the cyclodehydrogenation occur?
- 4) Have the authors tried the synthesis of Fe-Pcs by using NND precursor?
- 5) In the line 127 it is suggested that annealing at 500K desorbs PND molecules and it is compare with ref. 18, a study where >5 layers of AND precursor were deposited and lower temperatures of annealing are needed to obtain the polycyanine chains. How many layers of PND were deposited? Did annealing to temperatures between 300K and 500K produce changes?
- 6) Figure 3e shows the chemical structure of Fe-Pcs overlaid on a STM topography image and it looks like there is only one conformer possible to match it, with two right bright lobes in one direction and two bright left lobes in the perpendicular direction of the Pc. Is this the only conformer found for Fe-Pcs on Ag(111)?
- 7) About the Fe-Pcs synthesis on Ag(111) and Au(111) from PND precursor: Have the authors explored how the sample looks before annealing or annealing it at lower temperatures?
- 8) In the paragraph containing line 178, it is discussed that "domains with at least two different structures" are found after depositing PND on Au(111) at 300K. What is the ratio of each phase?
- 9) Figure 5b and 5c show the domains found after depositing PND on Au(111) and a close-up look of the mixed phase respectively. Could the authors include in the supplementary information the structure of the phase hexamer+tetramers or an STM image showing more than a unit cell of this structure?
- 10) Figure 5d shows different islands of Fe-Pcs on Au(111), from this image not all the islands have the same appearance. Could the authors include other close-up images of the islands to clarify this?
- 11) The annealing of PND on Au(111) with the presence of iron atoms reveals a material lost (Figure 5b and 5c), what is the coverage before and after? Does the yield of the cyclotetramerization depend on the phase the PND molecules are arranged before annealing? Does the yield of the cyclotetramerization depend on the amount of iron deposited?
- 12) In the line 187, the authors claim that the lower amount of byproducts around the Fe-Pcs islands on Au(111) than on Ag(111) is due to "the change in surface reactivity". But in line 243, the authors say that the deposition time (with the same flux) on Ag(111) and on Au(111) was different. Did the authors considered increasing the Fe deposition on Ag(111) or decreasing it on Au(111) to determine that it is the surface reactivity what produces the change in the amount of byproducts and it is not the availability of iron on the sample?
- 13) Starting in line 191 the authors describe a change in the appearance of Fe-Pcs on Au(111) after annealing up to 630K. Comparing Figure 5d with 5f, not only the appearance of the Fe-Pcs changes, also the coverage of the sample. Have the authors tried annealing at temperatures between 500 and 630K?
- 14) Regarding the cyclodehydrogenation of Fe-Pcs, have the authors observed differences in the C 1s XP spectra before and after the annealing at 630K?
- 15) The authors claim in line 67 that "For both systems, cyclodehydrogenation and thus flattening of the substituents seems

to be possible at higher annealing temperatures". In the manuscript the cyclodehydrogenation of Fe-Pcs synthesized from PND on Au(111) is shown, but it is not included if they have observed it for Fe-Pcs synthesized from PND on Au(111) nor for Ag-Pcs synthesized from NND on Ag(111). Could the authors include the evidences to claim that cyclodehydrogenation of both M-Pcs is possible?

16) In line 222 the authors conclude that "The smaller PND was found to undergo thermal desorption without reaction on Ag(111) and Au(111)", but the data about PND precursor on Au(111) annealed without the presence of iron atoms is missing from the manuscript. Could the authors include it?

Amelia Dominguez-Celorrio

Version 1:

Reviewer comments:

Reviewer #1

(Remarks to the Author)

The authors responded in detail and sufficiently to all my questions. In the result, the paper has been extended quite much, especially as concerns the supplementary information. Everything which could be explained in the scope of this paper has been included. Some remaining issues would demand further experiments and an extension of the methods used. So publication of the manuscript can be recommended.

Before publication a few smaller items should be corrected:

- Introduction; last sentence: reference should be to Figure S2, instead of S1.
- Results and Discussions: p.5; "Annealing of a PND ... The chains agglomerate, as can be seen in Figures 4d, e" (instead of Figure 4e).
- Results and Discussions: p.6; last section; "(cf. Figures S4 f, g for a close-up image)"
- Methods: "obtained from MaTeC with a surface misalignment of"
- References: the sign before the last author is sometimes "," and sometimes "&". There should be a unique notation.
- S 2.1: last sentence; "(cf. Figures 3, 6 and S6)" ? Figure S6 is without coadsorbed Fe.
- S 2.3: first line "and Figure 6 (on Au(111))"

Reviewer #2

(Remarks to the Author)

This manuscript explores the synthesis of phthalocyanines with extended π -electron systems from the perspectives of precursor molecule size and surface activity, with detailed experimental data. The revised version has improved significantly in quality by including additional experimental data and providing detailed responses to the reviewers' questions. I recommend publishing this revised manuscript in Communications Chemistry.

Reviewer #3

(Remarks to the Author)

Lukas J. Heuplick and co-workers have included in the reviewed version of the manuscript all the comments and suggestions I proposed in my review. The current version of the manuscript now includes new results, figures and sections. After a careful reading of the manuscript, I conclude that it shows interesting findings for the field and the results are properly discussed throughout the main text and in the supporting information when needed. I am now happy to accept the article for publication in Communications Chemistry, but here I include some final recommendations for the authors' consideration.

1. In the introduction the authors say, "As illustrated in Figure 1b, NND in the adsorbed state has three possible non-planar conformers with one tilted naphthyl unit, due to intramolecular steric hindrance." But throughout the manuscript they write about more than three conformers, maybe Figure 1 should include the four conformers (similar to the ones drawn in Figure S2) and then clarify that on the surface there are two enantiomers of each one.

2. The synthesis of PND is missing in the supporting information. Is it a commercially available precursor? Is the structure drawn in Figure 1c correct or is the precursor used a mixture of two possible enantiomers? In the latter case I would recommend drawing it simplified (without wedges).

3. I am not aware which one is the correct nomenclature of the NND compound but the authors use two different ones: 6,7-di(2-naphthyl)-2,3-naphthalenedicarbonitrile (main text) and 6,7-bis(2-naphthyl)naphthalene-2,3-dicarbonitrile (supporting information section 1.1.1).

4. The section "2.2. Detailed Analysis of Conformer Contributions" of the supporting information refers to metal-NPCs. I would suggest reorganizing it as it is confusing to start it with a first sentence about NND, then showing the analysis of PND derived metal-NPCs and then the analysis of NND derived metal-NPCs.

5. In the section 1.1.1 of the supporting information, when the authors say “yielding target compound 2”, do they mean NND instead of 2?

6. In page 6 of the main text, the authors say “formation of the polycyanine chains is reversible”. From my understanding, the process would be reversible if starting from Ag-NPc you could obtain polycyanine chains. As it is not the case, maybe using “reversible” to describe an intermediate product is not accurate.

7. In the conclusions, the authors should clarify the substrate they talk about. For example, when they say “Multilayers of PND at 450 K were found to form the open-chain covalent polycyanine, which are converted to regular naphthalocyanines at 550 K”, the reader could misinterpret that the formation of open-chain covalent polycyanine also occurs on Au(111) and the manuscript does not include results proving it.

8. In the page S4, there is an error when calling to Figure 2d,f, as it should be Figure S2d,f.

Response to the reviewers' comments

Reviewer #1

Comment: *The paper is recommended for publication, but minor revision is advised.*

The authors describe temperature-induced on-surface synthetic cyclotetramerization and metalation reactions of NND and PND towards metal-phthalocyanines, on noble metal (111) surfaces (Ag and Au, sometimes including co-adsorbed Fe atoms) studied by STM and XPS. In both cases, thermal annealing to 500 K has been applied to induce the process.

Response: We thank the reviewer for the careful reviewing of our manuscript and the helpful comments.

Comment: *After these annealing steps, seemingly only the yield of finalized molecules can be observed on the surface. Intermediate products as well as non-reacted precursors have been desorbed during annealing. Therefore arises the question whether shorter annealing times or somewhat lower annealing temperatures could possibly allow some further insight into the details of the ongoing reactions.*

Further, does the amount of co-adsorbed Fe atoms determine the yield of the metal-naphthalocyanines? Otherwise, one could expect to find single excess Fe atoms.

Response: Closer analysis shows that the temperatures steps used in this work indeed permit to capture intermediate states of the reaction. An example is added in the new Section 2.8 of the supplementary information, which shows the coexistence of the covalent cyclic tetramers (naphthalocyanines) and the metal-organic coordination network.

Actions taken: The following section was added to the supplementary information:

“2.8. Further STM Data of PND on Au(111)

Beside the structures shown in Figure 6 of the main text, PND on Au(111) features some less understood and, in this study, less thoroughly explored structural motifs. Figure S8 shows selected STM data and structural models. Figure S8a, b and c display the two observed self-assembled structures of PND on Au(111) as seen in the overview in Figure 6b and a proposed structural model (Figure S8d). Figure S8b shows a close-up of the tetramer-like motif, which is shown but not discussed in detail in the main text, with overlaid chemical models. Figure S8e and Figure S8g show STM images of the coordination network formed after adsorption of sub-stoichiometric amounts of Fe onto the pristine PND layer on Au(111) held at 300 K, before reaction at 500 K. In the overview image, one can see two different island motifs, closed-packed (Figure S8f) and row-like (Figure S8h). These two phases may result from different local molecule-to-iron ratios. However, while the close-up STM image (Figure S8g) shows a proposed chemical model, we were not able to resolve individual iron atoms and thus cannot inquire further.

After annealing this Fe/PND/Au(111) sample to 500 K, we observed the well-ordered Fe-NPc islands and another phase, both of which appear in the overview STM image in Figure 6d. Figure S8i shows a close-up of the other phase. The structure of this phase seems to closely resemble that of the iron-PND coordination network of the unreacted Fe/PND phase in Figure S8g. Therefore, we directly compare both phases (Figure S8j: mirrored and rotated cut-out of Figure S8i; Figure S8k: close-up image of the row-like network in Figure S8g). Despite the limited resolution, it can be concluded that Figure S8i shows a mirror domain of the iron-coordinated network shown in S8h. This implies that the reaction is not complete at 500 K, but results in the coexistence of the Fe-NPc product and the iron-PND coordination network. Figure S8l shows the STM image of a sample with higher iron-to-PND ratio (cf. Section 2.3) annealed to 550 K. The image shows immobile cyclic tetramers, which predominantly adsorb near the elbow sites. The other parts of the surface appear blurry, but the blurry features follow the herringbone reconstruction of the Au(111) surface. The molecular backbone appears with uniform brightness without pronounced protrusions, suggesting that the ligands have been flattened due to dehydrogenation reactions (as discussed in the main text). The flattening results in increased mobility resulting in the mostly blurry image. Only those of the flattened molecule which have engaged in intermolecular linking are sufficiently immobile for resolvable imaging under the applied experimental conditions.

Figure S8. (a) STM image of PND on Au(111) as deposited at 300 K, showing a larger area compared to Figure 6c. (b,c) Close-up STM images of the second PND structure formed of this sample with (b) and without (c) molecular overlay. An enlarged version of the overlaid molecular model is shown again in (d). (e) Overview STM image taken after deposition of a sub-stoichiometric amount of iron onto the PND layer. (f-h) Proposed structures as seen in the close-up image (g) without (f) and with (h) incorporation of iron atoms. (i) STM image of the row-like assembly of tetramer-type structures observed after annealing to 500 K. (j) Mirrored and rotated cut-out of (i) in contrast to the coordinated network from (g). (l) Overview STM image featuring very mobile species upon annealing a sub-monolayer PND with increased amounts of iron to 550 K (cf. Figure S3e and f). Scale bars: (a, l, j, k) 3 nm; (b, c) 2 nm; (e, l) 10 nm; (g) 4 nm. Tunneling parameters: (a) $U = 2.02$ V, $I = 0.09$ nA; (b, c) $U = -1.00$ V, $I = -0.18$ nA; (e) $U = -2.59$ V, $I = -0.23$ nA; (g) $U = -2.59$ V, $I = -0.16$ nA (i, j) $U = -0.63$ V, $I = -0.10$ nA; (k) $U = -2.59$ V, $I = -0.16$ nA; (l) $U = 2.51$ V, $I = 0.11$ nA. Bottom right: crystal orientation as determined in Figure S9.

Comment: Further, does the amount of co-adsorbed Fe atoms determine the yield of the metal-naphthalocyanines? Otherwise, one could expect to find single excess Fe atoms.

Response: Increasing the Fe-to-precursor ratio indeed changes the outcome of the reaction. Using an excess of Fe, we observe the formation of iron clusters which then serve as centers for the cyclooligomerization of the dinitriles, resulting in a mixture of various, difficult to characterize products of flower-shaped appearance (see e.g. Figure S3b,d below). However, a detailed systematic investigation of the influence of the Fe coverage is beyond the scope of this project.

Actions taken: We acquired additional data, which have been added to the supplementary information. The corresponding section 2.3 reads as follows:

“2.3. Changes to the Reaction due to Increased Fe-to-Molecule Ratios

In both PND/Fe samples shown in Figure 3 (on Ag(111)) and Figure 5 (on Au(111)), iron-to-molecule ratios below 1 Fe per 4 precursor molecules were discussed. This led to the desorption of excess PND molecules upon annealing, resulting in the stoichiometric ratios (1 Fe atom per 1 Npc molecule formed). Figure S3 shows results for samples with increased initial Fe-to-molecule ratios. On Ag(111), the ratios go up to over-stoichiometric amounts of Fe (Figure S3a,b after deposition at 300 K; Figure S3c,d after annealing to 500 K), while on Au(111), the ratios remain in the sub-stoichiometric regime (Figure S3e,f). As can be seen, the high Fe coverage on Ag(111) leads to a completely changed reaction behavior. Instead of regular fourfold symmetric features, various

irregular flower-shaped structures are obtained. Possibly, these are higher cyclic oligomers comprising five and more monomer units, with a cluster of Fe atoms in the center. Based on the available data, however, it is not possible to clarify the structure unambiguously.

On Au(111), increasing the amount of Fe to ratios still below the stoichiometric does not seem to affect the formation of the regular Fe-NPc, except that the edges of the Fe-NPc islands appear less straight at higher amounts of Fe.

Figure S3. (a, c) Large-scale image of a PND/Fe/Ag(111) sample with an over-stoichiometric amount of iron (≈ 5.8 Fe per 4 monomers; cf. Figure 3: ≈ 0.7 Fe per 4 monomers) deposited at 300 K and annealed to 500 K, respectively. (b, d) Close-up images corresponding to (a) and (c), respectively. (e) Large-scale image of a PND/Fe/Au(111) sample with a larger sub-stoichiometric amount of Fe (≈ 0.8 Fe per 4 monomers; cf. Figure 6: ≈ 0.3 Fe per 4 monomers), after annealing to 500 K. (f) Close-up image of the island in (e), where individual regular Fe-NPc molecules can be seen. Bottom right: crystal orientation as determined in Figure S9. Scale bars: (a, c, e) 10 nm; (b, f) 3 nm; (d) 1 nm. Tunneling parameters: (a) $U = -1.85$ V, $I = -0.11$ nA; (b) $U = -1.54$ V, $I = -0.11$ nA; (c) $U = -1.90$ V, $I = -0.13$ nA; (d) $U = -3.11$ V, $I = -0.16$ nA; (e) $U = -2.43$ V, $I = -0.14$ nA; (f) $U = -1.63$ V, $I = -0.13$ nA.”

Comment: *NND is expected to exhibit two isomers for each of the three conformers (Fig. 1) on the surface plane (having the tilted unit either on the left or the right side, respectively). For PND similarly two surface isomers are expected. Nevertheless, only one kind of them contributes to the final molecule (except for rarely encountered defects as shown in Fig. 2g). Are there observed complete mirror-counterparts in other regions of the substrate, or might the precursors molecules change mutually into the other isomer during annealing (e.g. by simultaneous hopping and twisting)?*

Response: We thank the reviewer for this question. Due to the limited statistics of our STM images it is difficult to give a clear answer based on experimental data alone. However, we added further data and analysis to the supplementary information (Section 2.2) addressing the question of the distribution of the conformers.

Actions taken: The following section was added to the supplementary information:

“2.2. Detailed Analysis of Conformer Contributions

In the two-dimensional (2D) confinement on the surface, NND can exhibit four different conformers, if only the relative orientation of the naphthyl substituents against each other is considered. However, due to the intrinsic tilt of the naphthyl substituents, there are two cases for each of these four conformers, depending on whether the right or the left substituent is tilted. Adsorption of the smaller PND can be analyzed in a similar way regarding its conformers. In the case of the intact PND precursor upon deposition at 300 K, the only island that could be resolved with sufficient detail (Figure 3b) show exclusively the “right side up” conformer. However, in absence of another image with sufficient sub-molecular resolution, we cannot prove or exclude that a corresponding mirror domain exists. Upon annealing PND in the presence of Fe to 500 K, we observed two different islands (Figure S2a and S2b) and can find at least 3 different Fe-NPc products, as illustrated in Figure S2c. Apparently, both PND conformers appear in the reacted Fe-NPc, with no clear preference for one of them.

Figure S2. (a,b) Close-up STM images of an island of PND-derived Fe-NPCs at a slightly different crystal positions, cf. also Figure 3e. (c) Observed conformers of Fe-NPC taking the tilt of the phenyl rings into account. (d, f) Two different NND-derived Ag-NPC ribbons at different crystal positions with and without molecular overlay. (In Figure 2g in the manuscript, a fraction of Figure S2d was overlaid with chemical models.) (e) Overview image of the NND precursor before annealing, demonstrating the presence of more than one island. (g) Contributions of the four different NND conformers in Figure 2d,f. The numbers give the observed number of conformers considering the tilting of the left or right naphthyl substituent. Scale bars: (a, b, d, f) 2 nm; (e) 40 nm. Tunneling parameters: (a) $U = -1.00$ V, $I = -0.23$ nA; (b) $U = -2.59$ V, $I = -0.14$ nA; (d) $U = -2.75$ V, $I = -0.10$ nA; (f) $U = -2.75$ V, $I = -0.08$ nA.

For NND, the above consideration leads to 8 different conformations of the monomer units and therefore to manifold combinations in the Ag-NPC chains (Figure S2d and S2f). For the intact NND precursor in Figure 2b in the manuscript, we were able to identify three different conformers in one noncovalent tetramer. Possibly, there are noncovalent tetramers which are composed of different conformers, but due to the lack of high-resolution data for more islands, this cannot be confirmed with certainty. However, already the overview image in Figure S2e suggests that not all assemblies might be equivalent. The center terrace shows a large island that is isolated from the step edges, while another island (bottom-left) extends up to the step edge. The two islands appear to be mirror domains, but are additionally rotated by an angle of $\sim 10^\circ$ relative to each other. However, we have no insight regarding the distribution of conformers in the two islands, or whether one conformation is more stable, or whether the conversion barrier is low enough for a statistical random distribution at room temperature.

Analyzing two different chains of naphthalocyanines obtained by annealing of NND, we find that there is not a singular most common monomer combination in the Ag-NPCs. By individually counting all four conformers, we find an approximately statistical ratio. The mixed (blue and purple) conformer occurs 19 times, the V-shaped (green) conformer occurs 10 times, and the parallel conformer (red) occurs 15 times. However, further considering the tilted side (cf. the numbers in Figure S2g) there seems to be a trend for a specific side in each conformer, which may be due to intermolecular and molecule-substrate interactions.”

Comment: In Figures 2, 3, and 5, the spatial orientation of the substrate surface should be indicated.

Response: We thank the reviewer for this suggestion. In all shown STM data, the substrate orientation is now indicated by the high symmetry axes above the scalebar. We added a section to the SI (Section 2.9) with the corresponding information about the directions were determined by STM.

Actions taken: The following section was added to the SI:

“2.9. Determination of the Crystal Orientation

Typically, we are not able to resolve the molecular overlayer and the substrate structure in the same image. Therefore, the substrate orientation was determined by the following approaches. Figure S9a shows an STM image of the Au(111) surface after desorption of PND with the herringbone reconstruction visible. The bright stripes of the reconstruction are oriented along the densely packed [11-2] direction (and equivalent directions). The blue lines thus show the orientation of the high symmetry directions. These orientations are used to indicate the substrate lattice for all samples prepared on Au(111). For Ag(111), a separately recorded image of the surface (Figure S9b) was used. Figure S9c shows a cut-out of the cleanly visible substrate and the determined orientation by green lines. For the PND experiments with multilayer coverage the usage of a different Ag(111) crystal was necessary. Figure S9d shows the corresponding substrate orientation for this crystal.

Figure S9. (a) STM image taken after desorbing PND from Au(111) (c.f. Figure S6). The herringbone reconstruction was used to determine the crystal orientation. (b) STM image of the used Ag(111) single crystal from a different preparation with atomically resolved substrate. (c) Cutout from (b) with determined crystal orientation d) STM image of the Ag(111) crystal used for the multilayer experiments of PND. Scale bars: (a) 30 nm; (b) 4 nm; (c, d) 2 nm. Tunneling parameters: (a) $U = -2.51$ V, $I = -0.10$ nA; (b, c) $U = -0.29$ V, $I = -0.14$ nA; (d) $U = -0.78$ V, $I = -0.18$ nA.”

Comment: There is shortly mentioned an additional phase visible in Figure 5b, top left. A close-up STM image of this phase should be added if possible (at least to the supplementary information).

Response: We thank the reviewer for this suggestion. We added the corresponding information in Section 2.8 of the SI. Figure S8b,c show a close-up of this island with a proposed structural model.

Actions taken: Section 2.8 was added to the SI.

Comment: In the references, there is sometimes a complete listing of all the authors otherwise an abbreviated list using “et al”. I would prefer to have always the complete list for “et al.” discriminates some of the coauthors, in my opinion.

Response: We thank the reviewer for catching this inconsistency.

Actions taken: All references in the revised version contain full author lists.

Reviewer #2

Comment: *Expanded phthalocyanines are a promising class of materials for optoelectronic applications. In this manuscript, the authors report the on-surface synthesis of two metallo-phthalocyanines Ag-NPc and Fe-NPc with extended electron systems from two different ortho-dicarbonitrile precursors studied by STM and XPS. The experimental results are interesting and important. The interpretation seem reasonable. The manuscript is well organized and written. However, there still are a few places that are not clear, which are listed below. If the authors could reasonably address these questions and comments, I would support the publication of this manuscript in the Communication chemistry.*

Response: We thank the reviewer for carefully reading our manuscript and the helpful advices.

Comment: 1. *Both the tetrameric structure of NND molecules on the Ag(111) surface and the formation of Ag-NPC reveal two benzene molecules that appear brighter. The author attributes this to the tilt of the groups. Why do they appear brighter after tilting? Is it due to higher spatial or charge effects?*

Response: Tilted phenyl groups have a higher relative height, thus are closer to the tip and, therefore, appear brighter, as seen for similar groups in the literature. To clarify this, we added the following sentence to the manuscript and supplied 2 additional references.

Actions taken: The following sentence was added on p. 3 of the revised manuscript:

“This tilting is typically found for compounds with similar rotationally flexible carbon-carbon single bonds in the literature and leads to an increased apparent height; examples include 5,10,15,20-tetraphenylporphyrin²⁰ or poly(para-phenylene)²¹.”

Comment: 2. *The author did not provide a coordination assembly structure induced by silver atoms from the assembly structure of NND molecules on the Ag(111) surface to the formation of the final polymer. Has the author not observed this coordination structure, or is it indeed not formed?*

Response: Indeed, we did not observe any coordination structures with Ag atoms. However, we cannot exclude that such structures exist under certain conditions outside the explored parameter space.

Comment: 3. *Reference 17 documented the creation of AuPC on the Au(111) surface using cyanide groups and surface Au atoms, tracing the detailed evolution from assembly structure to coordination structure, culminating in the final AuPc product. How did the author ascertain that the entity depicted in Figure 5 is Fe-NPC, considering it could potentially be Au-NPC? Has the author experimented with directly heating PND molecules on the Au(111) surface to explore the synthesis of AuNPC? Given that this paper primarily delves into investigating the impact of molecular configuration, substrate, and metal reactivity on phthalocyanine polymer synthesis, it should incorporate experimental data regarding NND on Au(111).*

Response: We thank the reviewer for this comment, which motivated us to perform additional experiments. For PND on Au(111), we observed the cyclotetramerization reaction only in presence of iron. Experiments with PND on Au(111) in the absence of iron resulted in desorption of the precursor without formation of NPc products (see Section 2.6 in the supplementary information).

However, contrasting observations were made for NND on Au(111). Annealing to 630 K resulted in a small fraction of cyclic tetramers, see the new Section 2.7 in the supplementary information. Some of the cyclic tetramers appear with a bright center, while others have a dark center, similarly as observed for the Ag complexes. It is concluded that the formation of Au-NPc on Au(111) is much less efficient than the formation of Ag-NPc on Ag(111), presumably due to the lower reactivity of Au.

Actions taken: The Sections 2.6 and 2.7 were added to the supplementary information.

“2.6. Desorption of PND from the Au(111) Surface

After annealing a sub-monolayer (0.61 ML) of PND on Au(111) to 500 K, the STM image (Figure S6a) show no longer any regular structures, but only small amounts or irregular residues. The absence of any regular structures suggests that there is no reaction of PND to Au-NPc or any chain-type structure. In line with this, the C 1s XPS (Figure S6b) shows a significantly decreased intensity (to 0.12 ML) and slight changes to the peak shape.

Figure S6. (a) Large scale image of 0.61 ML PND on Au(111) annealed to 500 K without co-adsorbed metal. (b) XP spectra showing the desorption of molecules upon annealing. Scale bar: (a) 30 nm; Tunneling parameters: (a) $U = -2.59$ V, $I = -0.10$ nA; Bottom right: crystal orientation as determined in Figure S9.

2.7. Reactivity of NND on Au(111) and Formation of Au-NPc

As discussed in the main text, the choice of substrate (Ag(111) or Au(111)) had only minor influence on the on-surface reactivity of PND with Fe. The only significant difference is in the shape of the island of the final Fe-NPc product. More substantial differences were observed for NND on the two surfaces regarding the formation of metallonaphthalocyanines. For comparison with the Ag(111) case discussed in the main text, we prepared a sub-monolayer of NND on Au(111) at 300 K. In corresponding STM images (Figure S7a,b), regular arrangements can be seen, however, due to the limited resolution, no exact molecular structure can be deduced. Annealing to 500 K results neither in desorption nor in an obvious reaction of the molecules (Figure S7c,d). Only by further increasing the annealing temperatures to 550 K and, thereafter, to 630 K, a small fraction of cyclic tetramers can be seen (Figure S7e,f and S7g,h, respectively). Some appear with a bright center, while others have a dark center, similarly as observed for the Ag complexes. It is concluded that the formation of Au-NPc on Au(111) is much less efficient than the formation of Ag-NPc on Ag(111), presumably due to the lower reactivity of Au.

Figure S7. (a, c, e, g) Large-scale images of a NND sample on Au(111) annealed to increasing temperatures: 300 K, 500 K, 550 K, 600 K. (b, d, f, h) Corresponding close-up images. (b) Upon deposition at 300 K some regular structures in a quasi-hexagonal lattice are observed. (d) Annealing to 500 K leads to random ordering of precursor shaped features. (f) Further annealing to 550 K leads to rarely encountered quasi fourfold symmetric features we attribute to coordination to a gold adatom. (h) Annealing to 600 K shows two phthalocyanine shaped features next to each other. However, in the center of the left one a bright protrusion is seen, which indicates the presence of a metal center. Scale bars: (a, c, e, g) 20 nm; (b) 4 nm; (d, f, h) 2 nm. Tunneling parameters: (a) $U = 2.43$ V, $I = 0.08$ nA; (b) $U = 2.43$ V, $I = 0.12$ nA; (c) $U = 1.79$ V, $I = 0.09$ nA; (d) $U = 1.79$ V, $I = 0.10$ nA (e) $U = -2.93$ V, $I = -0.10$ nA; (f) $U = -3.52$ V, $I = -0.14$ nA; (g) $U = -3.11$ V, $I = -0.17$ nA; (h) $U = -3.02$ V, $I = -0.14$ nA. Bottom right: crystal orientation as determined in Figure S9. “

Comment: 4. The “NND” on line 130 of page four should be “PND”.

Response: We thank the reviewer for pointing out this error.

Actions taken: The error was corrected.

Reviewer #3

Comment: *The manuscript of Lukas J. Heuplick and co-workers reports the on surface synthesis and characterization of two metallo phthalocyanines (M-Pcs) on Ag(111) and on Au(111), a topic where the authors have expertise and relevant previous publications. In this work they report the cyclotetramerization of 6,7-25 di(2-naphthyl)-2,3-naphthalenedicarbonitrile (NND) on Ag(111) leading to the silver naphthalocyanine, and the synthesis of a smaller iron naphthalocyanine on Au(111) and Ag(111) by cyclotetramerization of 6,7-diphenyl-2,3-naphthalenedicarbonitrile (PND) in the presence of co-adsorbed iron atoms. Throughout the manuscript, the authors discuss the formation of these metallo-phthalocyanines and their supramolecular arrangements supporting their claims with scanning tunnelling microscopy (STM) and X-Ray photoelectron spectroscopy (XPS) data. I consider the manuscript of interest to the field and I would recommend the work of Lukas J. Heuplick et al. to be published in Communications Chemistry after addressing the following points.*

Response: The reviewer's careful reading of our manuscript and the numerous valuable suggestions are highly appreciated.

Comment: 1. *In the line 76, it could be good to include the estimated number of ML deposited on Ag(111). Figure 2a suggests a complete ML, is that the case?*

Response: We thank the reviewer for pointing this out. We clarified this point by including the approximate coverage, as was determined by XPS using direct referencing to a saturation layer of a comparable molecule. The sample is not a saturated monolayer (1.0 ML), but corresponds to 0.40 ML.

Actions taken: We added the coverages for this instance and other instances in the manuscript and the SI.

Comment: 2. *In Figure 1 and throughout the manuscript, it is discussed that different conformers of the NND precursor coexist when adsorbed on a surface, but there is no mention of the stability of these three conformers on Ag(111). As it is the main reason for the termination of the non-covalent ribbons of Ag-Pcs, it could be interesting to include how many different conformers appear on the surface and an estimation of the % of each one after depositing NND precursor on Ag(111) and after the annealing up to 500K.*

Response: We thank the reviewer for this valuable comment. To address this question, we added a section to the SI discussing the prevalence of individual conformers as well as their distribution in the final M-NPCs (Section 2.2 and Figure S2). Since the tetramer apparently do not show a repeating pattern regarding the monomer conformation, we are not able to provide precise numbers or ratios.

Actions taken: The following section was added to the supplementary information file:

“2.2. Detailed Analysis of Conformer Contributions

In the two-dimensional (2D) confinement on the surface, NND can exhibit four different conformers, if only the relative orientation of the naphthyl substituents against each other is considered. However, due to the intrinsic tilt of the naphthyl substituents, there are two cases for each of these four conformers, depending on whether the right or the left substituent is tilted. Adsorption of the smaller PND can be analyzed in a similar way regarding its conformers. In the case of the intact PND precursor upon deposition at 300 K, the only island that could be resolved with sufficient detail (Figure 3b) show exclusively the “right side up” conformer. However, in absence of another image with sufficient sub-molecular resolution, we cannot prove or exclude that a corresponding mirror domain exists. Upon annealing PND in the presence of Fe to 500 K, we observed two different islands (Figure S2a and S2b) and can find at least 3 different Fe-NPC products, as illustrated in Figure S2c. Apparently, both PND conformers appear in the reacted Fe-NPC, with no clear preference for one of them.

Figure S2. (a,b) Close-up STM images of an island of PND-derived Fe-NPCs at a slightly different crystal positions, cf. also Figure 3e. (c) Observed conformers of Fe-NPC taking the tilt of the phenyl rings into account. (d, f) Two different NND-derived Ag-NPC ribbons at different crystal positions with and without molecular overlay. (In Figure 2g in the manuscript, a fraction of Figure S2d was overlaid with chemical models.) (e) Overview image of the NND precursor before annealing, demonstrating the presence of more than one island. (g) Contributions of the four different NND conformers in Figure 2d,f. The numbers give the observed number of conformers considering the tilting of the left or right naphthyl substituent. Scale bars: (a, b, d, f) 2 nm; (e) 40 nm. Tunneling parameters: (a) $U = -1.00$ V, $I = -0.23$ nA; (b) $U = -2.59$ V, $I = -0.14$ nA; (d) $U = -2.75$ V, $I = -0.10$ nA; (f) $U = -2.75$ V, $I = -0.08$ nA.

For NND, the above consideration leads to 8 different conformations of the monomer units and therefore to manifold combinations in the Ag-NPC chains (Figure S2d and S2f). For the intact NND precursor in Figure 2b in the manuscript, we were able to identify three different conformers in one noncovalent tetramer. Possibly, there are noncovalent tetramers which are composed of different conformers, but due to the lack of high-resolution data for more islands, this cannot be confirmed with certainty. However, already the overview image in Figure S2e suggests that not all assemblies might be equivalent. The center terrace shows a large island that is isolated from the step edges, while another island (bottom-left) extends up to the step edge. The two islands appear to be mirror domains, but are additionally rotated by an angle of $\sim 10^\circ$ relative to each other. However, we have no insight regarding the distribution of conformers in the two islands, or whether one conformation is more stable, or whether the conversion barrier is low enough for a statistical random distribution at room temperature.

Analyzing two different chains of naphthalocyanines obtained by annealing of NND, we find that there is not a singular most common monomer combination in the Ag-NPcs. By individually counting all four conformers, we find a approximately statistical ratio. The mixed (blue and purple) conformer occurs 19 times, the V-shaped (green) conformer occurs 10 times, and the parallel conformer (red) occurs 15 times. However, further considering the tilted side (cf. the numbers in Figure S2g) there seems to be a trend for a specific side in each conformer, which may be due to intermolecular and molecule-substrate interactions. “

Comment: 3. *About Ag-Pcs synthesised from NND precursor: Have the authors tried to anneal above 500K? Does the cyclodehydrogenation occur?*

Response: We thank the reviewer for this comment, which we addressed by additional experiments. We indeed observed flattened molecules due to intramolecular cyclodehydrogenation after annealing to 600 K. Our data suggest that the intramolecular cyclodehydrogenation is accompanied by intermolecular dehydrogenation reactions at this temperature.

Actions taken: We added the additional data to the supplementary information (Figure S1) and refer to it in the main manuscript.

“2.1. Further Data on NND/Ag(111): Cyclodehydrogenation at 630 K and Experiments with Co-Adsorbed Iron

Further annealing of the NND-based naphthalocyanine molecules on Ag(111), as shown Figure 2e, above 500 K is expected to induce intramolecular cyclodehydrogenation of the naphthyl substituents, resulting in flattening of the ligand. Indeed, after annealing to 630 K we no longer see molecules with regular bright protrusions at the periphery. In addition, all ordered structures disappeared (Figure S1a). The close-up image in Figure S1b confirms the presence cyclic tetramers with lobes of uniform brightness and a central protrusion. We attribute these structures to the flattened Ag-NPc in the Ag-up configuration. Besides the intramolecular reaction, also intermolecular dehydrogenative C-C coupling occurred, as can be seen in Figure S1b, which shows molecules that are linked at the periphery. These covalent aggregates can easily be imaged due to their low mobility, while the additional fuzzy features in Figure S1a,b suggest the presence of highly mobile molecules, which may be flat Ag-NPc molecules that did not undergo intermolecular coupling.

To get further understanding about the differences between NND and the smaller PND, we prepared an additional NND sample with co-adsorbed iron atoms. Figure S1c show the resulting structure after deposition at 300 K. Upon annealing to 500 K cyclic oligomers of irregular appearance are formed (Figure S1d). This result contrasts the regular cyclic tetramers (Fe naphthalocyanines) that were obtained by reaction of PND and Fe (cf. Figure 3, 6 and S6).

Figure S1. (a, b) Overview and close-up STM images of NND/Ag(111) upon annealing to 630 K. (c, d) STM images of NND/Ag(111) in presence of co-adsorbed iron after deposition at 300 K and annealing to 500 K, respectively. Scale bars: (a) 10 nm; (b) 2 nm; (c, d) 8 nm. Tunneling parameters: (a) $U = -2.59$ V, $I = -0.11$ nA; (b) $U = -2.59$ V, $I = -0.16$ nA; (c) $U = -1.74$ V, $I = -0.14$ nA; (d) $U = -2.15$ V, $I = -0.10$ nA. “

Comment: 4. *Have the authors tried the synthesis of Fe-Pcs by using NND precursor?*

Response: Yes, we tried this. In this case, we observed small cyclic oligomers with varying numbers of monomer units, similar to products obtained with over-stoichiometric amounts of iron. The regular 4 membered Fe-NPc, however, seems not to be a main product. We added this data to the supplementary information (Figure S1c,d, cf. reply to comment 3).

Comment: 5) *In the line 127 it is suggested that annealing at 500K desorbs PND molecules and it is compare with ref. 18, a study where >5 layers of AND precursor were deposited and lower temperatures of annealing are needed to obtain the polycyanine chains.*

How many layers of PND were deposited?

Did annealing to temperatures between 300K and 500K produce changes?

Response: Further analysis of the previous data revealed that PND coverage on Ag(111) remained in the sub-monolayer range. Consequently, we performed additional experiments with multilayer coverage, confirming the formation of polycyanine chains upon annealing to 450 K. These findings are now discussed in detail in the manuscript, and we have revised the corresponding sections of the abstract, introduction, and conclusion. Additional data have also been included in the supplementary information. We sincerely appreciate this comment, as it prompted us to re-examine our data, allowing us to provide a more in-depth contribution to the study of dinitrile oligomerization.

Actions taken: The following section and a new Figure 4 were added to the manuscript (pages 5-7). The corresponding smaller changes in the abstract, introduction, and conclusion are not listed here.

“Annealing of a PND multilayer (approx. 8×10^{14} molecules/cm²) on Ag(111) to 450 K without prior exposure to X-rays (cf. supplementary Figure S4) resulted in the formation of extended linear features, which are attributed to polycyanine chains (see the structure in Figure 4a), in line with previous work¹⁸. These chains are surrounded by smaller moieties (Figure 4b), which are most likely unreacted PND molecules. To desorb the residual unreacted PND, the sample was annealed to 470 K, resulting in a decrease of the total coverage from 0.52 ML to 0.39 ML, according to XPS. The STM image in Figure 4c, taken after the annealing, shows linear chains coexisting with islands containing cross-shaped fourfold symmetric molecules, which will be discussed later. Closer inspection of the chains reveals that each chain consists of three rows, a dark row in the center and two bright rows along the edges (Figure 4d and e). This matches well with the expected appearance of a polycyanine chain¹⁸, where the inner N-containing part binds more strongly to the substrate, whereas the edges are bent upward due to the non-planarity of the aromatic rings system. The chains agglomerate, as can be seen in Figure 4e, which is attributed to attractive interactions between the phenyl-substituents at the edges. To further confirm the formation of the covalent polycyanine chain (as opposed to a metal-organic or van-der Waals linked chains (cf. Figure S4 for further details), lateral tip manipulation of two different chains was performed (Figure 4f and 4k). By moving the tip along the indicated arrows in Figure 4h-j and 4l, we were able to bend an individual chain multiple times without breakage (Figure 4h-j), before a 90° rotation of the complete chains and further deformation of the chain occurred. By applying similar lateral movements to another, partly embedded chain (Figure 4l), bright protrusions and small dents appear at the bending positions. These protrusions were also seen previously for another polycyanine chain¹⁸ and was attributed to upright-standing chains segments due to compression at one edge of the chain.

The zoom-in STM image (Figure 4n) of the cross-shaped product reveals two distinct types of molecules, with and without a bright protrusion in the center. In line with the discussion for the NND cyclotetramerization above, we attribute these products to the regular M-NPcs, which, in the absence of other co-adsorbed metals, are assumed to incorporate Ag adatoms. The different contrasts at the centers are attributed to two different configurations, in which the Ag atom is either below the molecular plane (Ag-down, dark center, as in Figure 2) or above the molecular plane (Ag-up, bright protrusion), in agreement with previous work on Ag phthalocyanine on Ag(111).²² We propose that the presence of the organic multilayer during the formation of the covalent cyclic tetramers facilitates flipping movements of the Ag-NPc complex, resulting in a mixed phase of Ag-up and Ag down products.

The mixed phase of PND-based polycyanine chains and Ag-NPc was annealed to 550 K for 15 minutes to induce possible cyclodehydrogenation reactions of the phenyl substituents. After annealing, however, no chain-like structures were found on the surface (Figure 4o). This change is not due to desorption, as there was only a marginal reduction of the total coverage (from 0.39 ML to 0.36 ML, according to XPS). Instead, the coverage of Ag-NPc increased (cf. Figure S4 for a close-up image), indicating that formation of the polycyanine chains is reversible. The observed transformation suggests that the Ag-NPc represents the thermodynamically more stable product, while the polycyanine chains are the kinetically preferred product, in agreement with previous work on the ring-chain competition in on-surface oligomerizations.²⁴

Figure 4. (a) Reaction scheme for the formation of polycyanine chains and Ag-NPc cyclic covalent tetramers in PND multilayers at 450 K. (b) Large-scale STM image taken after annealing of a PND multilayer (approx. 8×10^{14} molecules/cm²) to 450 K on Ag(111). (c) STM image showing that further annealing of the sample in (b) to 470 K leads to desorption of residual monomers and the coexistence of chains and cyclic tetramers. (d) Close-up STM image of the polycyanine chains in (c). (e) Zoom-in STM image as indicated in (d) with overlaid chemical model. (f) Single chain perpendicular to a step edge before and parallel to the edge after (g) repeated lateral tip manipulations along the directions of the arrows in (h)-(j). (k) Different ensemble of chains. (l) Close-up STM image of the chain in (k) before manipulation. (m) Chain after tip manipulation showing a dent and additional bright protrusions attributed to upstanding chain segment. (n) Close-up STM image of the tetramer island in (c), showing cross-shaped cyclic covalent tetramers (NPcs) with and without bright central protrusions. (o) Large-scale STM image of the sample upon annealing to 550 K. Scale bars: (b, f, g, h, i, j, l, m) 5 nm; (c, k) 10 nm; (d) 3 nm; (e, n) 1 nm (o) 20 nm. Tunneling parameters: (b) $U = 1.20$ V, $I = 0.11$ nA; (c) $U = -1.28$ V, $I = -0.26$ nA; (d, e) $U = -1.40$ V, $I = -0.24$ nA; (f-i) $U = -2.59$ V, $I = -0.19$ nA; (j) $U = -3.02$ V, $I = -0.21$ nA; (k) $U = -2.15$ V, $I = -0.20$ nA; (l) $U = -2.15$ V, $I = -0.18$ nA; (m) $U = -2.15$ V, $I = -0.15$ nA; (n) $U = -0.20$ V, $I = -0.28$ nA; (o) $U = -3.31$ V, $I = -0.13$ nA."

Furthermore, the following related section 2.4 was added to the supplementary information:

"2.4. Further Data on the Polycyanine and Ag-NPc Formation of PND

In a first attempt to obtain polycyanine chains, we annealed a PND multilayer (>18 ML; no substrate signals observable in XPS) on Ag(111) to 450 K. Subsequent STM measurements (Figure S4a) reveal no ordered structures. We believe that the obtained residual coverage (1.2 ML, according to XPS) is due to cross-linking of the organic molecules in the multilayer caused by X-rays and secondary electrons during the XPS measurements of the multilayer after deposition (Figure S4e left). Therefore, for the next preparation, the data of which are

shown and discussed in the manuscript (cf. Figure 4), no XPS was measured before annealing. In this case, the coverage was determined by a quartz crystal microbalance.

Figure S4b-d show a close-up of the polycyanine chain that was obtained by annealing a PND multilayer that was not exposed to X-rays to 470 K. The same image is shown with three different overlays of hypothetical structures: the proposed polycyanine chain (Figure S4b) shows excellent agreement even for large numbers of repeat units; a coordination network of intact monomers with Ag adatoms and kinked bond angles (Figures S4c) fits similarly well along the chain, but is too large perpendicular to the chain direction; a metal-organic equivalent of the polycyanine chain with incorporated Ag atoms fits well in chain width, but is too long in the chain direction, as shown by the clear deviations already after few repeat units. This comparison provides further evidence for the formation of the covalent polycyanine chain.

Complementary XPS measurements (Figure S4e) support these findings. In the C 1s range of the PND multilayer XPS spectrum (Figure S4e, left), we find a large peak with a clearly separated shoulder to higher binding energies as discussed in detail for the sub-monolayers (Figure 5 in the manuscript). Upon annealing the intensity decreases significantly and a new peak appears at a lower binding energy.

XPS spectra taken after annealing the multilayer that was not exposed to X-rays to 450 K (Figure S4e right), we see a asymmetric peak. The asymmetry is attributed to the co-existence of reacted species and residual monomer molecules (as can be seen in Figure 4a). Upon further annealing to 470 K, the coverage decreases further (450 K: 0.51 ML; 470 K: 0.39 ML) and the peak becomes more symmetric. This is in line with the STM data, which show less monomers, while the chains and tetramers are the majority species. Further annealing to 550 K only leads to minor desorption (0.36 ML), supporting our observation of conversion of the chains into cyclic tetramers instead of decomposition and desorption.

To further identify the islands seen after annealing to 550 K we took close-up images of an islands (Figure S4f,g), where cyclic tetramers can clearly be identified. The molecules show bright central protrusions and are attributed to intact Ag-NPc in the Ag-up configuration.

Figure S4. (a) STM image of a PND multilayer sample (>18 ML; no substrate signal visible in XPS) after annealing to 450 K (1.2 ML). This multilayer was exposed to X-ray radiation prior to the annealing. (b-d) Polycyanine chain obtained by annealing a PND multilayer on Ag(111) without prior X-ray exposure (cf. Figure 4) with different hypothetical molecular overlays, as is discussed in the text. (e) XPS spectra of the C 1s range for both PND multilayer experiments with (left) and without (right) XPS measurement at 300 K before annealing to 450 K (coverage right: 450 K: 0.51 ML; 470 K: 0.39 ML; 550 K: 0.36 ML). (f, g) Close-up STM image of an island as in Figure 4o featuring tetramers without and with molecular overlay, respectively. Scale bars: (a) 20 nm; (b-d) 1 nm; (f, g) 2 nm. Tunneling parameters: (a) $U = 2.02$ V, $I = 0.14$ nA; (b-d) $U = -1.40$ V, $I = -0.24$ nA; (f, g) $U = -3.31$ V, $I = -0.13$ nA.”

Comment: 6. Figure 3e shows the chemical structure of Fe-Pcs overlaid on a STM topography image and it looks like there is only one conformer possible to match it, with two right bright lobes in one direction and two bright left lobes in the perpendicular direction of the Pc. Is this the only conformer found for Fe-Pcs on Ag(111)?

Response: We indeed see only one tetramer conformer in the shown island. However, in another island we find at least two different conformers. This observation we added to the SI (Figure S2 and related text, cf. response to comment 2).

Comment: 7. *About the Fe-Pcs synthesis on Ag(111) and Au(111) from PND precursor: Have the authors explored how the sample looks before annealing or annealing it at lower temperatures?*

Response: We studied the co-adsorbate for each substrate after deposition at 300 K. We added the data on Au(111) to the supplementary data (Figure S8 and related text). On Ag(111), we only studied a co-adsorbate with an over-stoichiometric amount of iron (cf. Figure S1 and S3). Other temperatures were not explored. This, however, may be considered in future projects.

Actions taken: The following Sections 2.8 and 2.3 were added to the supplementary information:

2.8. Further STM Data of PND on Au(111)

“Beside the structures shown in Figure 6 of the main text, PND on Au(111) features some less understood and, in this study, less thoroughly explored structural motifs. Figure S8 shows selected STM data and structural models. Figure S8a, b and c display the two observed self-assembled structures of PND on Au(111) as seen in the overview in Figure 6b and a proposed structural model (Figure S8d). Figure S8b shows a close-up of the tetramer-like motif, which is shown but not discussed in detail in the main text, with overlaid chemical models. Figure S8e and Figure S8g show STM images of the coordination network formed after adsorption of sub-stoichiometric amounts of Fe onto the pristine PND layer on Au(111) held at 300 K, before reaction at 500 K. In the overview image, one can see two different island motifs, closed-packed (Figure S8f) and row-like (Figure S8h). These two phases may result from different local molecule-to-iron ratios. However, while the close-up STM image (Figure S8g) shows a proposed chemical model, we were not able to resolve individual iron atoms and thus cannot inquire further.

After annealing this Fe/PND/Au(111) sample to 500 K, we observed the well-ordered Fe-NPc islands and another phase, both of which appear in the overview STM image in Figure 6d. Figure S8i shows a close-up of the other phase. The structure of this phase seems to closely resemble that of the iron-PND coordination network of the unreacted Fe/PND phase in Figure S8g. Therefore, we directly compare both phases (Figure S8j: mirrored and rotated cut-out of Figure S8i; Figure S8k: close-up image of the row-like network in Figure S8g). Despite the limited resolution, it can be concluded that Figure S8i shows a mirror domain of the iron-coordinated network shown in S8h. This implies that the reaction is not complete at 500 K, but results in the coexistence of the Fe-NPc product and the iron-PND coordination network. Figure S8l shows the STM image of a sample with higher iron-to-PND ratio (cf. Section 2.3) annealed to 550 K. The image shows immobile cyclic tetramers, which predominantly adsorb near the elbow sites. The other parts of the surface appear blurry, but the blurry features follow the herringbone reconstruction of the Au(111) surface. The molecular backbone appears with uniform brightness without pronounced protrusions, suggesting that the ligands have been flattened due to dehydrogenation reactions (as discussed in the main text). The flattening results in increased mobility resulting in the mostly blurry image. Only those of the flattened molecule which have engaged in intermolecular linking are sufficiently immobile for resolvable imaging under the applied experimental conditions.

Figure S8. (a) STM image of PND on Au(111) as deposited at 300 K, showing a larger area compared to Figure 6c. (b,c) Close-up STM images of the second PND structure formed of this sample with (b) and without (c) molecular overlay. An enlarged version of the overlaid molecular model is shown again in (d). (e) Overview STM image taken after deposition of a sub-stoichiometric amount of iron onto the PND layer. (f-h) Proposed structures as seen in the close-up image (g) without (f) and with (h) incorporation of iron atoms. (i) STM image of the row-like assembly of tetramer-type structures observed after annealing to 500 K. (j) Mirrored and rotated cut-out of (i) in contrast to the coordinated network from (g). (l) Overview STM image featuring very mobile species upon annealing a sub-monolayer PND with increased amounts of iron to 550 K (cf. Figure S3e and f). Scale bars: (a, l, j, k) 3 nm; (b, c) 2 nm; (e, l) 10 nm; (g) 4 nm. Tunneling parameters: (a) $U = 2.02$ V, $I = 0.09$ nA; (b, c) $U = -1.00$ V, $I = -0.18$ nA; (e) $U = -2.59$ V, $I = -0.23$ nA; (g) $U = -2.59$ V, $I = -0.16$ nA (i, j) $U = -0.63$ V, $I = -0.10$ nA; (k) $U = -2.59$ V, $I = -0.16$ nA; (l) $U = 2.51$ V, $I = 0.11$ nA. Bottom right: crystal orientation as determined in Figure S9. “

“2.3. Changes to the Reaction due to Increased Fe-to-Molecule Ratios

In both PND/Fe samples shown in Figure 3 (on Ag(111)) and Figure 5 (on Au(111)), iron-to-molecule ratios below 1 Fe per 4 precursor molecules were discussed. This led to the desorption of excess PND molecules upon annealing, resulting in the stoichiometric ratios (1 Fe atom per 1 NPc molecule formed). Figure S3 shows results for samples with increased initial Fe-to-molecule ratios. On Ag(111), the ratios go up to over-stoichiometric amounts of Fe (Figure S3a,b after deposition at 300 K; Figure S3c,d after annealing to 500 K), while on Au(111), the ratios remain in the sub-stoichiometric regime (Figure S3e,f). As can be seen, the high Fe coverage on Ag(111) leads to a completely changed reaction behavior. Instead of regular fourfold symmetric features, various irregular flower-shaped structures are obtained. Possibly, these are higher cyclic oligomers comprising five and more monomer units, with a cluster of Fe atoms in the center. Based on the available data, however, it is not possible to clarify the structure unambiguously.

On Au(111), increasing the amount of Fe to ratios still below the stoichiometric does not seem to affect the formation of the regular Fe-NPc, except that the edges of the Fe-NPc islands appear less straight at higher amounts of Fe.

Figure S3. (a, c) Large-scale image of a PND/Fe/Ag(111) sample with an over-stoichiometric amount of iron (≈ 5.8 Fe per 4 monomers; cf. Figure 3: ≈ 0.7 Fe per 4 monomers) deposited at 300 K and annealed to 500 K, respectively. (b, d) Close-up images corresponding to (a) and (c), respectively. (e) Large-scale image of a PND/Fe/Au(111) sample with a larger sub-stoichiometric amount of Fe (≈ 0.8 Fe per 4 monomers; cf. Figure 6: ≈ 0.3 Fe per 4 monomers), after annealing to 500 K. (f) Close-up image of the island in (e), where individual regular Fe-NPc molecules can be seen. Bottom right: crystal orientation as determined in Figure S9. Scale bars: (a, c, e) 10 nm; (b, f) 3 nm; (d) 1 nm. Tunneling parameters: (a) $U = -1.85$ V, $I = -0.11$ nA; (b) $U = -1.54$ V, $I = -0.11$ nA; (c) $U = -1.90$ V, $I = -0.13$ nA; (d) $U = -3.11$ V, $I = -0.16$ nA; (e) $U = -2.43$ V, $I = -0.14$ nA; (f) $U = -1.63$ V, $I = -0.13$ nA.”

Comment: 8. In the paragraph containing line 178, it is discussed that “domains with at least two different structures” are found after depositing PND on Au(111) at 300K. What is the ratio of each phase?

Response: We found approximately similar fractions of the two structures, but no in-depth analysis has been performed.

Comment: 9. Figure 5b and 5c show the domains found after depositing PND on Au(111) and a close-up look of the mixed phase respectively. Could the authors include in the supplementary information the structure of the phase hexamer+tetramers or an STM image showing more than a unit cell of this structure?

Response: We thank the reviewer for the suggestion to provide more data addressing this point.

Action taken: Additional data were added to the supplementary information (Figure S8, cf. response to comment 7).

Comment: 10. Figure 5d shows different islands of Fe-Pcs on Au(111), from this image not all the islands have the same appearance. Could the authors include other close-up images of the islands to clarify this?

Response: We analyzed close-up images of this phase in detail in the supplementary information (Figure S8 and related text, cf. also response to comment 7). Our main finding is that the second phase is an organometallic network containing unreacted monomers.

Comment: 11. The annealing of PND on Au(111) with the presence of iron atoms reveals a material lost (Figure 5b and 5c), what is the coverage before and after? Does the yield of the cyclotetramerization depend on the phase the PND molecules are arranged before annealing? Does the yield of the cyclotetramerization depend on the amount of iron deposited?

Response: The coverage of PND on Au(111) decreases from 0.67 ML at 300 K to 0.34 ML at 500 K. Since the coverage of Fe atoms remains constant, the iron-to-molecule ratio increases from ≈ 0.3 to ≈ 1.0 Fe atoms per 4 monomers due to the PND desorption. Therefore, it appears reasonable to conclude that the coverage loss is due to desorption of unreacted monomers. We added a section with further discussion of XPS data in the

supplementary information (Figure S5 and related text) and report the values of the iron-to-molecule ratios in Section 2.3 (cf. response to comment 7).

Due to instrumental constraints, we are not able to study the exact same position before and after the annealing procedure with STM. The reason for this is that the sample must be removed from the STM during the annealing and it is not possible to find the same spot on the surface again after reintroducing the sample. Therefore, we cannot investigate the phase dependency further on an experimental basis.

The yield does indeed depend on the amount of iron. Over-stoichiometric amounts of Fe lead to decreased reaction control and higher cyclic oligomers (cf. Section 2.3 in the supplementary information).

Actions taken: The following Section 2.5 was added to the supplementary information:

“2.5. Additional C 1s XPS Spectra

In Figure 5 in the manuscript, only a selection of XPS spectra is shown for the sake of clarity. However, taking a closer look at further C 1s spectra in Figure S5 provides additional insight. Already upon deposition of Fe onto PND on both surfaces, the peak shape changes substantially. Especially the shoulder towards higher binding energies decreases, which is probably a consequence of the coordination of the nitrile groups to iron atoms. However, since there is a sub-stoichiometric amount of iron (cf. Figure S3 and S8), uncoordinated nitrile groups are present as well. After annealing to 500 K, the coverage is reduced in all cases. For PND, this is easily understood taking the amount of iron into consideration. At 500 K there is roughly 1 Fe left for each tetramer, while the residual monomers desorb. For NND, the explanation might be similar, i.e., the residual amount may depend on the Ag adatom concentration.

Further annealing at 630 K slightly decreases the coverage in the cases of PND/Fe/Au(111) (Figure S5a) and NND/Ag(111) (Figure S5c). For PND/Fe/Au(111), we attribute this to desorption of unreacted monomers (cf. Section 2.8).

In addition, we observe small changes in the peak width upon the annealing from 500 K to 630 K (increase of FWHM: Figure S5a, 0.98 eV to 1.05 eV; Figure S5b, 1.07 eV to 1.18 eV; Figure S5c, 1.21 eV to 1.30 eV), which are attributed to increasing inhomogeneity due to the coexistence of individual flat M-NPc molecules and already interconnected irregular networks as shown for NND (cf. Figure S1a and b).

Figure S5. XP Spectra of the C 1s regime for (a) PND/Fe/Au(111), (b) PND/Fe/Ag(111) and (c) NND/Ag(111). Shown are for each the spectra taken after deposition of the precursors at 300 K, after reaction at 500 K and further annealing to 630 K. For the smaller PND additionally the co-adsorbate with iron is shown. Coverages: (a) 300 K: 0.67 ML, 500 K: 0.34 ML, 630 K: 0.26 ML; (b) 300 K: 0.52 ML, 500 K: 0.33 ML, 630 K: 0.30 ML; (c) 300 K: 0.76 ML, 500 K: 0.40 ML, 630 K: 0.30 ML. “

Comment: 12) In the line 187, the authors claim that the lower amount of byproducts around the Fe-Pcs islands on Au(111) than on Ag(111) is due to “the change in surface reactivity”. But in line 243, the authors say that the deposition time (with the same flux) on Ag(111) and on Au(111) was different. Did the authors considered increasing the Fe deposition on Ag(111) or decreasing it on Au(111) to determine that it is the surface reactivity what produces the change in the amount of byproducts and it is not the availability of iron on the sample?

Response: We believe that this comment is based on a misunderstanding. We proposed that the different surface reactivity of Ag(111) and Au(111) is the reason for straighter edges of the Fe-NPc on Au(111). The edge shape

should be the result between intermolecular interaction and molecule-substrate interaction. Therefore, on Ag(111) where the latter has a higher contribution deviations of the energetically favored structure with straight edges can easier be stabilized. To clarify this, we removed the misleading comment about byproducts.

Actions taken: The manuscript was modified as follows (p. 9):

“Note that the edges of the islands are straighter than on the Ag(111) surface. This can be explained by differences in the adsorbate-substrate interaction. Due to the weaker interaction with the Au(111) surface, the intermolecular interactions are comparatively more significant. Deviations from the edge structure with the lowest energy (i.e., straight edges) can therefore be stabilized on Ag(111), but less so on Au(111), resulting in more well-defined edges on Au(111).”

Comment: 13. Starting in line 191 the authors describe a change in the appearance of Fe-Pcs on Au(111) after annealing up to 630K. Comparing Figure 5d with 5f, not only the appearance of the Fe-Pcs changes, also the coverage of the sample. Have the authors tried annealing at temperatures between 500 and 630K?

Response: The coverage decreases from 0.34 ML at 500 K to 0.26 ML at 630 K, in line with our interpretation of the second phase seen in the STM data as unreacted species. The unreacted precursor molecules most likely desorb upon further annealing before they can react. The coverage decrease is mentioned in the supplementary information (Figure S5 and related text; cf. comment 11). We indeed studied another sample annealed to 550 K. At this temperature, mostly mobile products were obtained, as is discussed in more detail in the SI (Figure S8I; cf. response to comment 7).

Comment: 14. Regarding the cyclodehydrogenation of Fe-Pcs, have the authors observed differences in the C 1s XP spectra before and after the annealing at 630K?

Response: C 1s XPS data are discussed in more detail in the revised SI (Figure S5 and related text, cf. also response to comment 11). Indeed, small changes of the peak width, but not of the peak position, are observed.

Comment: 15. The authors claim in line 67 that “For both systems, cyclodehydrogenation and thus flattening of the substituents seems to be possible at higher annealing temperatures”. In the manuscript the cyclodehydrogenation of Fe-Pcs synthesized from PND on Au(111) is shown, but it is not included if they have observed it for Fe-Pcs synthesized from PND on Au(111) nor for Ag-Pcs synthesized from NND on Ag(111). Could the authors include the evidences to claim that cyclodehydrogenation of both M-Pcs is possible?

Response: We thank the reviewer for pointing out this omission. We annealed NND on Ag(111) to 630 K and observed flattening of the molecules.

Actions taken: The corresponding data have been added to the SI (Figure S1 and related text; cf. also response to comment 3).

Comment: 16. In line 222 the authors conclude that “The smaller PND was found to undergo thermal desorption without reaction on Ag(111) and Au(111)”, but the data about PND precursor on Au(111) annealed without the presence of iron atoms is missing from the manuscript. Could the authors include it?

Response: We added an STM image after the annealing step and complementary XPS data to the SI (Figure S6).

Actions taken: The following section was added to the supplementary information:

“2.6. Desorption of PND from the Au(111) Surface

After annealing a sub-monolayer (0.61 ML) of PND on Au(111) to 500 K, the STM image (Figure S6a) show no longer any regular structures, but only small amounts or irregular residues. The absence of any regular structures suggests that there is no reaction of PND to Au-NPc or any chain-type structure. In line with this, the C 1s XPS (Figure S6b) shows a significantly decreased intensity (to 0.12 ML) and slight changes to the peak shape.

Figure S6. (a) Large scale image of 0.61 ML PND on Au(111) annealed to 500 K without co-adsorbed metal. (b) XP spectra showing the desorption of molecules upon annealing. Scale bar: (a) 30 nm; Tunneling parameters: (a) $U = -2.59$ V, $I = -0.10$ nA; Bottom right: crystal orientation as determined in Figure S9.”

Response to the reviewers' comments

Reviewer #1

Comment: *The authors responded in detail and sufficiently to all my questions. In the result, the paper has been extended quite much, especially as concerns the supplementary information. Everything which could be explained in the scope of this paper has been included. Some remaining issues would demand further experiments and an extension of the methods used. So publication of the manuscript can be recommended.*

Before publication a few smaller items should be corrected:

Response: We thank the reviewer for the careful reading of our revisions and the final remarks, which resulted in further improvement of our manuscript.

Comment: *Introduction; last sentence: reference should be to Figure S2, instead of S1.*

Response: We believe that Figure S1a,b is indeed correct. We refer to the cyclodehydrogenation reaction as the final step of our reaction scheme. This is proven to be possible for NND in Figure S1, while Figure S2 shows further analysis of the different conformer and enantiomer contributions.

Comment: *Results and Discussions: p.5; "Annealing of a PND ... The chains agglomerate, as can be seen in Figures 4d, e" (instead of Figure 4e).*

Response: We thank the reviewer for this suggestion.

Actions taken: We added this information in the following sentence.

"The chains agglomerate, as can be seen in Figure 4d and e, which is attributed to attractive interactions between the phenyl-substituents at the edges."

Comment: *Results and Discussions: p.6; last section; "(cf. Figures S4 f, g for a close-up image)"*

Response: We thank the reviewer for pointing this out.

Actions taken: We modified this sentence as seen below.

Instead, the coverage of Ag-NPc increased (cf. Figure S4f and g for a close-up image), indicating a follow-up degradation reaction of the polycyanine chains.

Comment: *Methods: "obtained from MaTeC with a surface misalignment of"*

Response: We thank the reviewer for catching this error.

Actions taken: We changed the sentence accordingly.

Comment: *References: the sign before the last author is sometimes "," and sometimes "&". There should be a unique notation.*

Response: We ensured that now "&" is always the separator for the last author.

Comment: *S 2.1: last sentence; "(cf. Figures 3, 6 and S6)" ? Figure S6 is without coadsorbed Fe.*

Response: We thank the reviewer pointing out this erroneous figure reference.

Actions taken: We removed the reference to Figure S6.

"This result contrasts the regular cyclic tetramers (Fe naphthalocyanines) that were obtained by reaction of PND and Fe (cf. Figure 3 and 6)."

Comment: *S 2.3: first line "and Figure 6 (on Au(111))"*

Response: We thank the reviewer for catching this error.

Actions taken: We changed the sentence accordingly.

"In both PND/Fe samples shown in Figure 3 (on Ag(111)) and Figure 6 (on Au(111)), iron-to-molecule ratios below 1 Fe per 4 precursor molecules were discussed."

Reviewer #2:

Comment: This manuscript explores the synthesis of phthalocyanines with extended π -electron systems from the perspectives of precursor molecule size and surface activity, with detailed experimental data. The revised version has improved significantly in quality by including additional experimental data and providing detailed responses to the reviewers' questions. I recommend publishing this revised manuscript in Communications Chemistry.

Response: We are grateful to the reviewer for their positive evaluation of our manuscript and recommending the publication.

Reviewer #3:

Comment: Lukas J. Heuplick and co-workers have included in the reviewed version of the manuscript all the comments and suggestions I proposed in my review. The current version of the manuscript now includes new results, figures and sections. After a careful reading of the manuscript, I conclude that it shows interesting findings for the field and the results are properly discussed throughout the main text and in the supporting information when needed. I am now happy to accept the article for publication in Communications Chemistry, but here I include some final recommendations for the authors' consideration.

Response: We are grateful to the reviewer for the careful reading of our manuscript, the additional comments, and their recommendation for publication.

Comment: 1. In the introduction the authors say, "As illustrated in Figure 1b, NND in the adsorbed state has three possible non-planar conformers with one tilted naphthyl unit, due to intramolecular steric hindrance." But throughout the manuscript they write about more than three conformers, maybe Figure 1 should include the four conformers (similar to the ones drawn in Figure S2) and then clarify that on the surface there are two enantiomers of each one.

Response: We thank the reviewer for pointing out this inconsistency. We indeed have to consider 4 conformers in the 2D confinement and corrected this.

Actions taken: We corrected the following sentence.

"As illustrated in Figure 1b, NND in the adsorbed state has four possible non-planar conformers with one tilted naphthyl unit, due to intramolecular steric hindrance."

In addition, Figure 1 was modified and the additional conformer was added as follows:

Comment: 2. The synthesis of PND is missing in the supporting information. Is it a commercially available precursor? Is the structure drawn in Figure 1c correct or is the precursor used a mixture of two possible enantiomers? In the latter case I would recommend drawing it simplified (without wedges).

Response: We thank the reviewer for catching this. PND is not commercially available, but was synthesized by us as well. Regarding the second point, the free PND is of course achiral and only becomes chiral when adsorbed (and thus immobilized) on the surface at low temperatures. In this case, there are in principle indeed two enantiomers. However, the focus is not on this. The wedges mainly indicate that the phenyl rings are not coplanar with the surface, which is similar to the NND case. Therefore, for consistency we believe it is better to keep the wedges.

Actions taken: We added the PND synthesis details to the supporting information (Section 1.1.1. Preparation of 6,7-diphenyl-2,3-naphthalenedicarbonitrile).

Comment: 3. I am not aware which one is the correct nomenclature of the NND compound but the authors use

two different ones: 6,7-di(2-naphthyl)-2,3-naphthalenedicarbonitrile (main text) and 6,7-bis(2-naphthyl)naphthalene-2,3-dicarbonitrile (supporting information section 1.1.1).

Response: We thank the reviewer for pointing out this inconsistency.

Actions taken: We changed the name in the synthesis details to keep the consistency.

“1.1.2. Preparation of 6,7-di(2-naphthyl)-2,3-naphthalenedicarbonitrile”

Comment: 4. The section “2.2. Detailed Analysis of Conformer Contributions” of the supporting information refers to metal-NPCs. I would suggest reorganizing it as it is confusing to start it with a first sentence about NND, then showing the analysis of PND derived metal-NPCs and then the analysis of NND derived metal-NPCs.

Response: We thank the reviewer for this suggestion, which further enhances our manuscript.

Actions taken: We restructured the beginning of section S2.2 as followed.

“In the two-dimensional (2D) confinement on the surface, PND can exhibit one conformer and NND can exhibit four conformers, if only the relative orientation of the aryl substituents relative to each other is considered. However, due to the intrinsic tilt of the substituents, there are two cases for each conformer, depending on whether the right or the left substituent is tilted. In the case of the intact PND precursor upon deposition at 300 K, the only island that could be resolved with sufficient detail (Figure 3b) shows exclusively the “right side up” conformer.”

Comment: 5. In the section 1.1.1 of the supporting information, when the authors say “yielding target compound 2”, do they mean NND instead of 2?

Response: We thank the reviewer for catching this small error. This should, indeed, be NND.

Actions taken: We changed the sentence accordingly.

Comment: 6. In page 6 of the main text, the authors say “formation of the polycyanine chains is reversible”. From my understanding, the process would be reversible if starting from Ag-NPC you could obtain polycyanine chains. As it is not the case, maybe using “reversible” to describe an intermediate product is not accurate.

Response: We thank the reviewer for pointing this out. Indeed, we see that our current sentence is misleading. “Reversible” was supposed to refer to the chains’ carbon-nitrogen bonds, which need to break and form again to switch between “chain” and “ring”. However, the reviewer is completely right that the final product is not equal to the initial precursor, and therefore complete reversibility is not present. To clarify this point, we changed the corresponding sentence.

Actions taken: We changed the sentence for clarification.

“Instead, the coverage of Ag-NPC increased (cf. Figure S4f and g for a close-up image), indicating a follow-up degradation reaction of the polycyanine chains, resulting in the formation of Ag-NPC.”

Comment: 7. In the conclusions, the authors should clarify the substrate they talk about. For example, when they say “Multilayers of PND at 450 K were found to form the open-chain covalent polycyanine, which are converted to regular naphthalocyanines at 550 K”, the reader could misinterpret that the formation of open-chain covalent polycyanine also occurs on Au(111) and the manuscript does not include results proving it.

Response: We thank the reviewer for catching these misleading paragraphs. Indeed, we currently have no experimental data regarding formation of a polycyanine on Au(111).

Actions taken: We now additionally mention the crystal in the following sentences.

“Multilayers of PND on Ag(111) at 450 K were found to form the open-chain covalent polycyanine, which are converted to regular naphthalocyanines at 550 K.”

“While submonolayers of PND were found to desorb on both Au(111) and Ag(111) without engaging in reactions, the reactive cyclotetramerization of sub-monolayer PND was induced by co-adsorbed Fe atoms.”

Comment: 8. In the page S4, there is an error when calling to Figure 2d,f, as it should be Figure S2d,f.

Response: We thank the reviewer for pointing this out.

Actions taken: We changed the sentence accordingly.